# Hydrodynamic regimes modulate nitrogen fixation and the mode of diazotrophy in Lake Tanganyika

Benedikt Ehrenfels [1,2], Kathrin B. L. Baumann[1,2], Robert Niederdorfer [1], Athanasio S. Mbonde[3], Ismael A. Kimirei [3,4], Thomas Kuhn[5], Paul M. Magyar[5], Daniel Odermatt [1], Carsten J. Schubert [1,2], Helmut Bürgmann [1], Moritz F. Lehmann [5], Bernhard Wehrli [1,2] & Cameron M. Callbeck [1,5] ✉

The factors that govern the geographical distribution of nitrogen fixation are fundamental to providing accurate nitrogen budgets in aquatic environments. Model-based insights have demonstrated that regional hydrodynamics strongly impact nitrogen fixation. However, the mechanisms establishing this physical-biological coupling have yet to be constrained in field surveys. Here, we examine the distribution of nitrogen fixation in Lake Tanganyika – a model system with well-defined hydrodynamic regimes. We report that nitrogen fixation is five times higher under stratified than under upwelling conditions. Under stratified conditions, the limited resupply of inorganic nitrogen to surface waters, combined with greater light penetration, promotes the activity of bloom-forming photoautotrophic diazotrophs. In contrast, upwelling conditions support predominantly heterotrophic diazotrophs, which are uniquely suited to chemotactic foraging in a more dynamic nutrient landscape. We suggest that these hydrodynamic regimes (stratification versus mixing) play an important role in governing both the rates and the mode of nitrogen fixation.

Biological nitrogen (N) fixation, or the conversion of inert $N_2$ gas into bioavailable forms of N, is an important process in aquatic environments where it may relieve N limitation of primary productivity. Nitrogen fixation itself may be regulated by the availability of phosphorus (P), iron (Fe), and/or fixed N[1–6]. While phosphorus is an essential macronutrient for the growth of all organisms, Fe is critical due to the high Fe demand of the key enzyme responsible for $N_2$ fixation, nitrogenase. The availability of fixed N, such as nitrate and ammonium, generally slows down $N_2$ fixation but does not necessarily deactivate it[6–12]. In rare cases, $N_2$ fixation rates even correlate positively with nitrate concentrations[13,14]. The supply and interplay of these nutrients (P, Fe, and fixed N) to and in aquatic environments, potentially leads to the large-scale patterns that are currently observed. The stratified,

oligotrophic open oceans, such as the subtropical ocean gyres, are typically important sites of $N_2$ fixation, driving up to 50% of organic N and carbon export rates[15]. However, absolute rates of $N_2$ fixation in these regions can vary by orders of magnitude[16], and are possibly linked to varying fluxes of P and Fe into these ecosystems[17]. Furthermore, the $N_2$-fixing communities can differ from cyanobacteria-dominated regions, such as the tropical North Atlantic[18], to regions where non-cyanobacteria may be the more important contributors[19].

In contrast to the nutrient-poor surface waters of the open oceans, eastern boundary upwelling systems frequently introduce excess P to surface waters (and potentially Fe[20]) that have been proposed to foster $N_2$ fixation[21]. Field studies, however, have found upwelling $N_2$ fixation rates to be overall lower than in much of the open

[1]Eawag, Swiss Federal Institute of Aquatic Science and Technology, Department Surface Waters – Research and Management, Kastanienbaum, Switzerland. [2]ETH Zurich, Institute of Biogeochemistry and Pollutant Dynamics, Zurich, Switzerland. [3]TAFIRI, Tanzania Fisheries Research Institute, Kigoma, Tanzania. [4]TAFIRI, Tanzania Fisheries Research Institute, Dar es Salaam, Tanzania. [5]University of Basel, Department of Environmental Sciences, Basel, Switzerland. ✉e-mail: cameron.callbeck@unibas.ch

ocean (e.g. Selden et al.[22], and references therein). While rates vary widely from below the limit of detection to as high as 100 nM d$^{-1}$ [13,15,22–30], they do not re-supply the N lost before/during upwelling as predicted by previous biogeochemical models[21]. More recent estimates indicate that N$_2$ fixation contributes little as a new N source in upwelling regions, contributing only ~1% to the export of organic N and carbon[15]. Nevertheless, heterotrophic diazotrophs are ubiquitously found in nutrient-rich waters – including the more recently studied coastal and polar systems – which are often but not exclusively dominated by non-cyanobacterial diazotrophs[23,24,31–36]. Despite their prevalence in pelagic environments[31,37], heterotrophic diazotrophs may contribute to low rates (<1 nM d$^{-1}$) of N$_2$ fixation[37,38].

The large spatial variability in both the rates of N$_2$ fixation and the associated diazotrophic communities, even in environments with similar biogeochemical conditions, indicate a lack of a mechanistic understanding[39]. A clear causal link between nutrient supply, largely driven by hydrodynamics (e.g. stratification or upwelling), and N$_2$ fixation rates as well as the ambient N-fixing microbial communities has not been established[8–10,40]. Yet, recent models highlight that the parameterization of physical-biological processes is fundamental to understanding the observed distribution of N$_2$ fixation in marine settings[15].

The sheer size of the ocean and the complexity of hydrodynamic processes within also further challenge our capacity to identify mechanisms of physical-biological coupling. To this end, Lake Tanganyika (the second deepest lake in the world), which exhibits contrasting hydrodynamic regimes between its north and south basins, represents an ideal testing ground for studying how upwelling and stratification may regulate N$_2$ fixation and the community of diazotrophs. The lake water column is generally depleted in N relative to P and is characteristically structured into oligotrophic surface waters, a nitrate-rich intermediate zone, and a euxinic hypolimnion[41]. During the dry season, the south basin experiences a strong wind-driven upwelling, while the north/central basins remain permanently stratified[42]. Previous studies have reported non-diazotrophic phytoplankton blooms in the south during the dry season, whereas massive blooms of filamentous N$_2$-fixing cyanobacteria typically emerge during more intensely stratified periods[43,44]. A microbial community survey has identified additional putative photoautotrophic and heterotrophic diazotrophs[45].

Here, we explored the distribution of N$_2$ fixation and associated diazotrophs under contrasting hydrodynamic regimes across a north-south transect in Lake Tanganyika (Fig. 1a). Using a combination of stable isotope applications and metagenomics, we find that the hydrodynamic regimes selected for particular ecophysiological traits that might enable the key diazotrophs to thrive under either stratified or upwelling conditions.

## Results and discussion

### Hydrodynamic regimes in Lake Tanganyika

We sampled Lake Tanganyika during the transition from the rainy to the dry season in April/May 2018. This period was characterized by lake-wide stratification with warm surface waters (up to 27.6 °C) overlaying cooler waters (Fig. 1b). Under these stratified conditions, the water column was deficient in oxygen (<10 μM oxygen) below ~110–120 m in the north and below ~130–140 m in the south (Fig. 1b and Supplementary Fig. 1a). The deeper oxycline in the south is a robust feature of Lake Tanganyika[46], and develops as a result of deep vertical mixing down to ~150 m[42]. Even though the entire lake was stratified at the time of sampling, north-south differences were apparent. For instance, the thermocline, separating the warm surface layer from deeper waters, was generally positioned at a much shallower depth (<23 m) in the southern basin compared to most stations sampled in the northern and central basins (45–64 m). The shallower thermocline in the south arose as a consequence of wind-driven upwelling induced by the southeast trade winds[42].

Overall, in the upper 50 m of the lake, nitrate concentrations were mostly below the limit of detection (<0.25 μM in 63 % of samples), except for Station 1 and at one depth at Station 3 (Fig. 1c, i). Below ~50 m, nitrate concentrations increased sharply, reaching a maximum of 8–9 μM near 100 m water depth. Nitrate concentrations eventually fell below the detection limit from ~150 m downward, due to anammox and denitrification activity[47] (see also Supplementary Discussion). By contrast, soluble reactive phosphate in the surface waters was below the limit of detection ( < 0.19 μM) only at stations 1 and 2 (Fig. 1d, j), and increased from 50 m downward, reaching up to 1.3–1.5 μM near the nitrate maximum. Ammonium concentrations were below the limit of detection (<0.26 μM) in oxygenated waters and increased below ~150 m (Supplementary Fig. 1b). The low N:P ratios, which were below the Redfield ratio of 16:1 in 95 % of all samples (Supplementary Fig. 1c), indicating that soluble reactive phosphate was available in excess over DIN throughout the water column, consistent with previous studies[43,44]. The resupply of DIN to the euphotic zone (delimited at a surface irradiance cutoff of 1%; Fig. 1) is controlled by the primary thermocline[44], separating the nitrate-depleted surface zone from nitrate-rich intermediate waters. At Stations 2–5, located in the heavily stratified north/center, the thermocline was situated 9 m below the lower boundary of the euphotic zone (Fig. 1h), thereby diminishing the resupply of nitrate to phytoplankton. An additional surface thermocline, observed at Stations 3 and 4, further limited the vertical nutrient transport to the surface mixed layer (upper ~30 m; Fig. 1c). In contrast, at Stations 8 and 9 in the south, the thermocline was situated 10 m above the lower boundary of the euphotic zone, which allows nitrate to be resupplied for assimilation in surface waters more efficiently[44]. Estimated fluxes of nitrate into the euphotic zone ranged from 140–300 μmol m$^{-2}$ d$^{-1}$ in the north/center, and up to 690 μmol m$^{-2}$ d$^{-1}$ at Station 9 in the south (Fig. 1i).

Lake Tanganyika also exhibited strong latitudinal trends with regard to chlorophyll, an indicator of primary productivity. In the south, the primary chlorophyll maximum (PCM) was located between the thermocline and the bottom of the euphotic zone, whereas the PCM occurred near the surface in the north and central basins (Fig. 1e). In-situ chlorophyll fluorescence was highest at Stations 7–9 in the south (>1.5 μg L$^{-1}$; Fig. 1e, k), consistent with previously described basin-wide patterns[48]. In further support, higher CO$_2$ fixation rates and $^{13}$C-enriched particulate organic carbon (indicating higher fractional dissolved CO$_2$ consumption) were reported for the south, compared to the north[49]. We ascribe the greater surface productivity in the south to the nutrient upwelling; accordingly, we relate the less productive surface PCM in the north to a reduced turbulent diffusive DIN supply due to stronger stratification. Furthermore, the lower surface productivity in the north enabled greater light penetration to deeper depths that (Supplementary Fig. 2), in turn, sustained deep blooms of anoxygenic phototrophs that resulted in an anoxic chlorophyll maximum (ACM) occurring at 140–170 m (Fig. 1e)[47]. This, combined with the fact that vertical mixing causes a deepening of the oxycline could additionally constrain the available niche for anoxygenic phototrophic sulfur bacteria. The presence of an ACM therefore serves as an indicator of permanently stratified waters and is otherwise absent in the more productive south (Supplementary Fig. 3).

### Upwelling impacts filamentous cyanobacteria and associated nitrogen fixation

Based on our $^{15-15}$N$_2$ incubation experiments, we find the greatest rates of N$_2$ fixation in the euphotic zone (i.e., above 50 m depth), ranging from below the limit of detection to 22.9 nM N d$^{-1}$ (Fig. 1f). Our maximum volumetric rates were higher than those measured in the neighboring Lake Malawi (~0.5–5 nM N d$^{-1}$)[50], but comparable to maximum rates measured in the oligotrophic Laurentian Great Lakes of 30 nM N d$^{-1}$ (with a whole lake mean value between 1–4 nM N d$^{-1}$;[14]). It should be

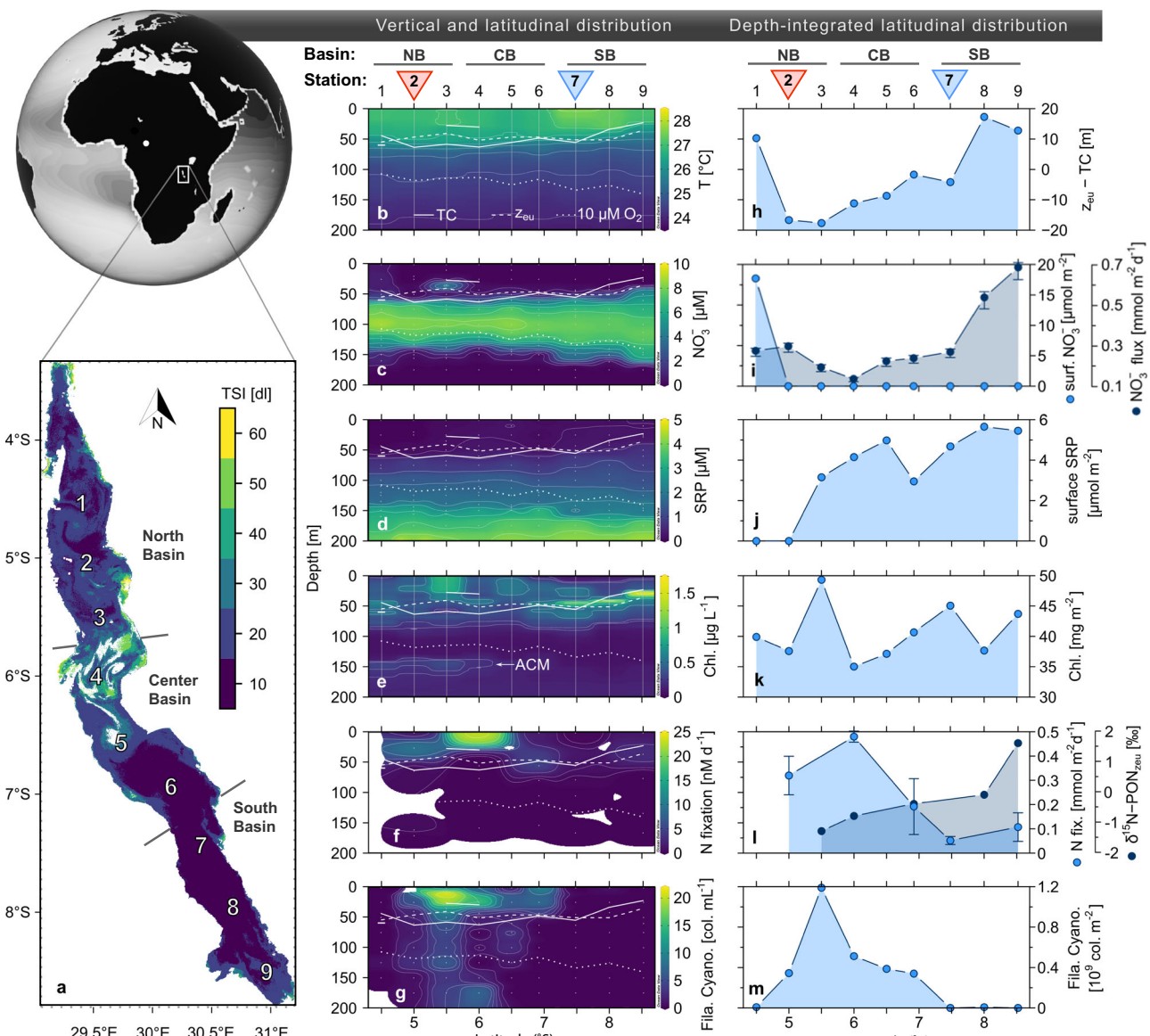

**Fig. 1 | Distribution of nutrients, chlorophyll, filamentous nitrogen-fixing cyanobacteria, and rates of nitrogen fixation across a north-south transect in Lake Tanganyika.** The transect and profiles shown were taken in April/May 2018. **a** Aerial surface chlorophyll-a concentrations are represented by the Trophic State Index (TSI), obtained from Sentinel satellite data from 27 April to 7 May 2018. TSI values are proxies of the trophic state: 0–30, indicates clear oligotrophic waters; 40–50, mesotrophic; 60–70, eutrophic (see methods). Panels **b**–**g** show the vertical distributions of temperature, nitrate, soluble reactive phosphate (SRP), chlorophyll (Chl), volumetric rates of nitrogen fixation, and abundance of filamentous cyanobacteria. Note that dots indicate discrete sampling depths, while solid vertical lines mark CTD profiles. In panels **h**–**m**, depth-integrated values and general north-south patterns are depicted. **h** Indicates the position of the thermocline (TC) relative to the euphotic depth ($Z_{eu}$); **i** surface nitrate inventory (0–25 m) and nitrate flux into the euphotic zone; **j** surface SRP inventory (0–25 m); **k** depth-integrated in-situ chlorophyll fluorescence (0–125 m); **l** depth-integrated values of nitrogen fixation (0–175 m) and the mean $\delta^{15}N$ of particulate organic nitrogen in the euphotic zone (0–43 m; $\delta^{15}N$-PON$_{zeu}$); **m** depth-integrated values (0–125 m) of filamentous cyanobacteria. Error bars represent standard errors. The main sampling Stations 2 and 7 are marked by red and blue downward triangles, respectively. Stations 1–3, 4–6, and 7–9 represent the North (NB), Center (CB), and South basins (SB), respectively. Data in panels **i** and **l** are presented as mean values from duplicate samples (distinct samples), with error bars representing the standard error. The world globe, temperature and chlorophyll-a presented in panels a, b, and e, respectively, were published previously[44,47] (licensed under a Creative Commons Attribution 4.0: https://creativecommons.org/licenses/by/4.0/).

noted that in small eutrophic lakes rates of $N_2$ fixation can be 1–2 orders of magnitude greater[51,52]. Moreover, our rates were higher than in most marine upwelling areas, where $N_2$ fixation typically ranges from below detection to ~10 nM N d$^{-1}$ [8,10,13,23], although exceptional rates above 100 nM N d$^{-1}$ have been reported[9,32]. Along our north-south transect, rates of $N_2$ fixation were greatest in the north/central basins, with maximum rates ranging from 7.5–22.9 nM N d$^{-1}$ (Stations 2, 4, and 6), whereas rates never exceeded 3 nM N d$^{-1}$ in the south (Stations 7 and 9; Fig. 1f). In accordance with the volumetric rates, the depth-integrated $N_2$ fixation estimates ranged between 192–480 µmol N m$^{-2}$ d$^{-1}$ in the north/

central basins, which was between 2- to 10-fold higher than values reported in the south basin (46–107 µmol N m$^{-2}$ d$^{-1}$; Fig. 1l).

Averaged on a basin scale, $N_2$ fixation in the upper 50 m added 300 ± 70, and 60 ± 40 µmol N m$^{-2}$ d$^{-1}$ to the north/central and south basins, respectively (Table 1). The estimated vertical nitrate flux into the euphotic zone amounted to 230 ± 30, and 500 ± 10 µmol N m$^{-2}$ d$^{-1}$ in the north/central and south basins, respectively. Atmospheric deposition plus river input may add another 160 and 140 µmol N m$^{-2}$ d$^{-1}$, based on annual estimates for the two regions[53]. These numbers indicate that $N_2$ fixation was the main source of bioavailable N fueling new

**Table 1 | Nitrogen fluxes with standard errors (mmol N m$^{-2}$ d$^{-1}$) in Lake Tanganyika during April/May 2018 for the north/central and south basins**

|  |  | North/Center | South | Reference |
|---|---|---|---|---|
| Euphotic zone | N fixation | 0.30 ± 0.07 (*n* = 3) | 0.06 ± 0.04 (*n* = 2) | this study |
|  | NO$_3^-$ flux | 0.23 ± 0.03 (*n* = 6) | 0.50 ± 0.10 (*n* = 3) | this study |
|  | Deposition | 0.13 | 0.13 | 53 |
|  | Rivers | 0.02 | 0.003 | 53 |
|  | Total new production | 0.68 | 0.69 | – |
| Anoxic zone | N fixation | 0.02 ± 0.00 (*n* = 3) | 0.001 ± 0.00 (*n* = 2) | this study |
|  | NO$_3^-$ flux | 0.85 ± 0.20 (*n* = 6) | 0.64 ± 0.16 (*n* = 3) | this study |
|  | NH$_4^+$ flux | 0.48 ± 0.04 (*n* = 6) | 0.71 ± 0.40 (*n* = 3) | this study |
|  | N removal | 2.53 ± 0.32 (*n* = 5) | 0.91 ± 0.20 (*n* = 3) | this study |

Atmospheric deposition and riverine input were extracted from Langenberg et al.[53]. The mean values presented for the North/Center and South basins are based on Stations 1–6 and 7–9, respectively. The sample size (*n*) is indicated.

production in the north and central parts of Lake Tanganyika, whereas the vertical nitrate flux was the major source of bioavailable N in the south.

Our $^{15}$N natural abundance analysis of particulate organic N (PON), which provides integrated information on N$_2$ fixation into phytoplankton biomass over longer time scales, showed strong vertical and north-south trends that were consistent with rates determined from short-term $^{15\text{-}15}$N$_2$ incubation experiments (Fig. 1l). The vertical distribution of δ$^{15}$N-PON indicated consistently low values in the euphotic zone (0–43 m) attaining a mean (based on all stations) of −0.3‰, indicative of low-δ$^{15}$N N$_2$ as an important N source sustaining the phytoplankton community (Supplementary Fig. 4)[54–56]. By contrast, below 50 m depth, the higher average δ$^{15}$N-PON values of 1.4‰, could be ascribed to the assimilation of nitrate (-1.9‰) as the main N source rather than by N$_2$ fixation. In line with the high N$_2$ fixation rates in the north, we find the lowest δ$^{15}$N-PON$_{Zeu}$ signature at Station 3 (northernmost analyzed station) of −1.4‰. In contrast, a significantly higher δ$^{15}$N-PON$_{Zeu}$ value of 1.6‰ was observed in the southern basin at Station 9, while intermediate values were observed for stations in between ranging from −0.8 to −0.1‰. Together the δ$^{15}$N-PON$_{Zeu}$ north-south trend corroborates the differences in surface N$_2$ fixation across the lake.

The north and central basins of Lake Tanganyika were also a hotspot for diazotrophic filamentous cyanobacteria (Stations 2–6; Fig. 1g, m), reaching up to 22 colonies mL$^{-1}$, whereas the south was largely void of filamentous cyanobacteria. In our parallel study addressing the phytoplankton composition, we found that these blooms were mainly comprised of known diazotrophs *Dolichospermum* (>99 %), and to a small extent *Anabaenopsis* (<1 %)[44,57]. The role of *Dolichospermum* in N$_2$ fixation was further supported by the presence of a *nifHDK* operon identified in a recovered Tanganyika *Dolichospermum* metagenome-assembled genome (MAG) discussed below. Moreover, our sampled *Dolichospermum* colonies contained up to four heterocyst cells, which are specialized in the catalysis of N$_2$ fixation[44].

Depth-integrated abundances of filamentous cyanobacteria compared versus rates of euphotic zone N$_2$ fixation and δ$^{15}$N-PON were significantly correlated (Spearman, *p* < 0.05; Fig. 1l, m and Supplementary Fig. 9b, f). Together this suggested that *Dolichospermum* was a key contributor to the elevated N$_2$ fixation rates in the northern and central basins (Fig. 1f, l). Based on this clear relationship (Supplementary Fig. 9b), and the average number of cells per colony[44], we estimate that single-cell N$_2$ fixation rates were roughly 15 fmol N cell$^{-1}$ d$^{-1}$, which is within the range of values reported for *Dolichospermum* in the Baltic Sea (-1–220 fmol N cell$^{-1}$ d$^{-1}$)[57]. Filamentous cyanobacteria were also shown to be responsible for the highest rates of N$_2$ fixation in other large lakes[14,50], as well as in many oligotrophic regions of the ocean[58–60].

Across the lake, the abundance of filamentous cyanobacteria exhibited a significant negative correlation with the nitrate flux from subsurface waters (Spearman, $R^2$ = 0.79, *p* < 0.05; Supplementary Fig. 9a). Moreover, the rates of N$_2$ fixation showed a trend towards higher activity with a decreasing nitrate flux, albeit this trend was relatively noisy ($R^2$ = 0.50, *p* < 0.2; Supplementary Fig. 9g). Nonsteady state conditions might potentially contribute to a wider margin of error in correlative analyses at some stations. For example, a fluctuating DIN supply is expected for Stations 1 and 2, which are positioned near the river inlets of Rusizi and Malagarasi, respectively. These rivers experience large DIN fluctuations due to rain events during the transition from the wet to the dry season[53]. Therefore, N$_2$ fixation rates may suffer from a lagged response by the diazotroph community toward a fluctuating nitrate supply[44]. Nevertheless, based on the δ$^{15}$N-PON$_{Zeu}$ signature (a long-term indicator of N$_2$ fixation) we find a strong correlation with the nitrate flux ($R^2$ = 0.79; *p* < 0.05; Supplementary Fig. 9d). In further support, the correlation of the nitrate flux with the abundances of filamentous cyanobacteria (Supplementary Fig. 9a), combined with their correlation to N$_2$ fixation activity (Supplementary Fig. 9b), affirm that the nitrate flux is a key moderating factor of cyanobacterial N$_2$ fixation in Lake Tanganyika.

Thus, we attribute the high N$_2$ fixation rates by filamentous cyanobacteria in the north and center basins of Lake Tanganyika to the low resupply of DIN, which provides a competitive advantage to N$_2$-fixing phytoplankton[44,61]. An exception to this trend is the northernmost Station 1, which was largely devoid of N$_2$-fixing cyanobacteria (Fig. 1g, m). Here, the relatively high availability of DIN in surface waters (Fig. 1c, i), which may have been injected from deeper layers by internal waves[62], or may stem from riverine sources[53], likely negated the competitive advantage of N$_2$-fixing phytoplankton as discussed elsewhere[44]. Likewise, in the south basin, the wind-driven upwelling of nutrient-rich waters impeded the proliferation of filamentous cyanobacteria. Instead, these waters were dominated by heterotrophic diazotrophs (discussed below) and were associated with much lower N$_2$ fixation rates compared to the north basin. Interestingly, interseasonal data also support these findings, as blooms of filamentous cyanobacteria tend to be confined to the stratified conditions observed during seasonal transitions as well as the rainy season. So far, there are no reports of blooms occurring during the dry season upwelling period[43,63]. Likewise, in similarly large lakes[50] and marine environments[9,59,64], high N$_2$ fixation rates related to free-living filamentous cyanobacteria appear to be most often associated with stratified rather than upwelling conditions. Correspondingly, modeling studies have found that diazotrophs have a niche in nutrient-poor stratified ocean regions[15], especially when surface waters lack DIN but P and Fe are available (Fig. 1i, j)[7,65].

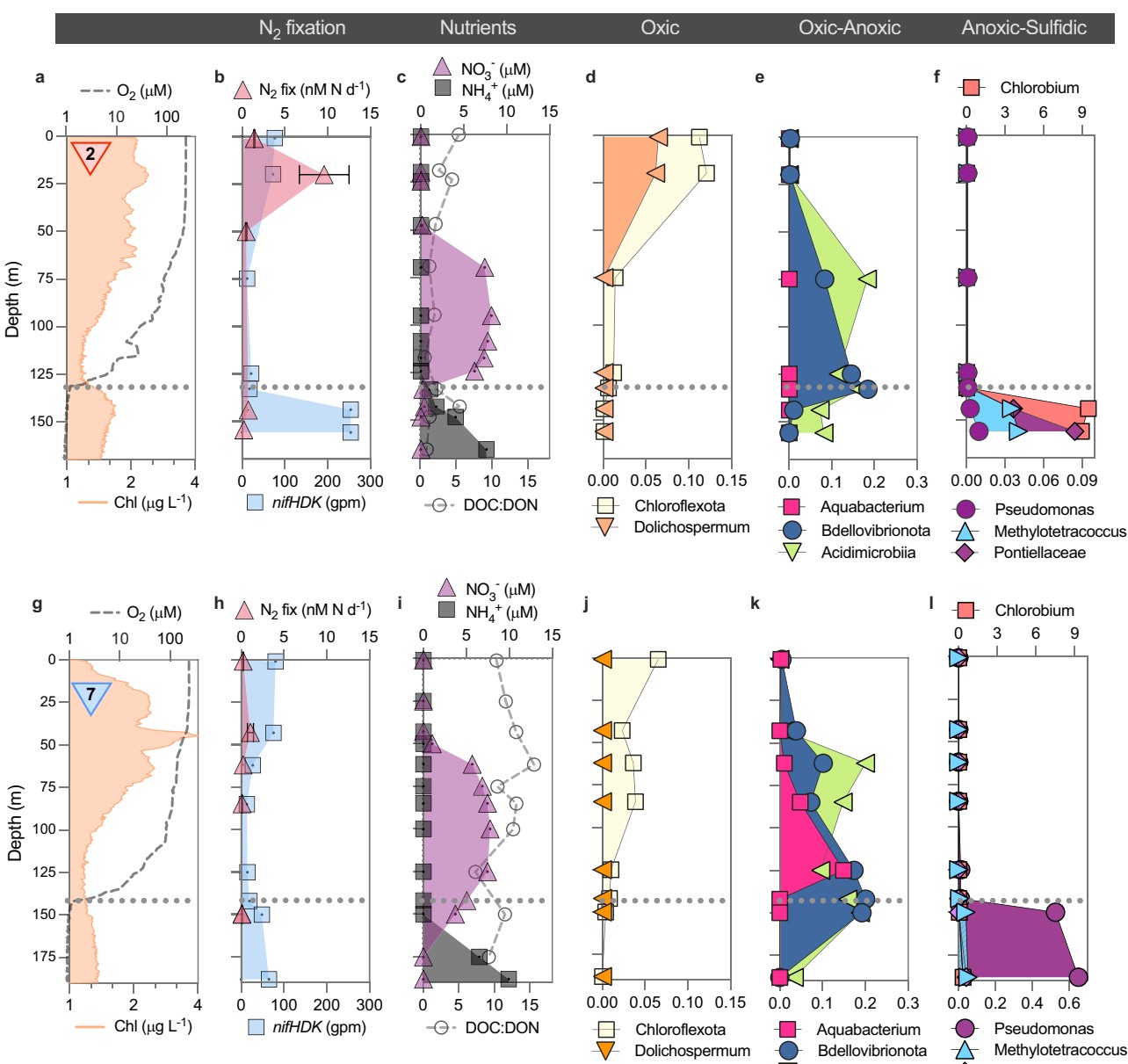

**Fig. 2 | Niche partitioning of N₂ fixing microbes identified in Lake Tanganyika.**
**a–f** Station 2 is situated in the northern basin, while (**g–l**) Station 7 is located in the southern basin, and is also marked by red and blue downward triangles, respectively. Shown are the vertical distribution profiles of chlorophyll (based on in-situ fluorescence), oxygen, nutrients, rates of N₂ fixation, *nifHDK* gene abundances (recovered from metagenomes; Supplementary Table 1), and the relative abundance of diazotroph candidate metagenome-assembled genomes. Note that the concentration data for chlorophyll, oxygen, and DOC:DON ratio as well as N₂ fixation rates are log-transformed. The upper limit of the anoxic zone is delimited at an O₂ cutoff of 1 μM (gray dotted line). Data in panels **b** and **h** are presented as mean values for duplicate samples (distinct samples), with error bars representing the standard deviation. DOC dissolved organic carbon, DON dissolved organic nitrogen.

## Nitrogen fixation in the oxygen-deficient zone

We also observed rates of N₂ fixation in the oxygen-deficient waters of Lake Tanganyika. Specifically, the greatest deep N₂ fixation rates were reported in the anoxic waters coinciding with the ACM at Station 2 in the north, with rates reaching up to 0.75 nM N d⁻¹ at 144 m (Figs. 1f and 2a, b). In the south, which harbors no ACM (Figs. 1e and 2g, h), N₂ fixation rates were almost an order of magnitude lower, with a maximum of 0.11 nM N d⁻¹ measured at Station 7. Overall, our range of N₂ fixation rates in anoxic waters (0.11- 0.75 nM N d⁻¹) measured in Lake Tanganyika is in the same order of magnitude as determined in anoxic waters of other stratified lakes or basins (0.44–1.01 nM N d⁻¹;[37,66,67]) and marine oxygen minimum zones[23,68,69]. However, a potential caveat is that the nominal pore size of glass fiber filters (0.7 μm) used for N₂

fixation measurements is larger than some planktonic bacteria. Thus, N₂ fixation by smaller diazotrophs, which are characteristic of anoxic depths, may be underestimated[70]. Moreover, we cannot completely exclude that trace oxygen contamination may have inhibited N₂ fixation by strict anaerobes, such as *Chlorobium*, which dominates in the ACM (discussed below). Therefore, the rates measured in the anoxic zone likely represent the lower limit of deep/anoxic N₂ fixation activity in Lake Tanganyika.

The measurable rates of N₂ fixation in the anoxic zone raised the possibility of a co-occurrence between N₂ fixation and N removal (Supplementary Fig. 7; Supplementary Discussion), which has been previously observed in other stratified lakes and basins[37,67]. While such a co-occurrence may exist, rates of N₂ fixation in the anoxic zone

satisfied at most 1% of the fixed N required to balance N loss (Table 1). Instead, the export of newly-fixed N from filamentous cyanobacteria in surface waters could supply up to 12% of the fixed N for anammox/denitrification in the northern oxygen-deficient zone (Table 1). In addition, depth-integrated rates of $N_2$ fixation and N loss were positively correlated (Spearman, $R^2 = 0.7$, $p < 0.05$; Supplementary Fig. 8), suggesting that the export of newly-fixed N from filamentous cyanobacteria in surface water could have contributed to the enhanced rates of N loss in the oxygen-deficient zone reported in the northern basin[47].

### Ecophysiology of diazotrophs under contrasting hydrodynamic regimes

Our analysis of the vertical distribution of the nitrogenase genes *nifHDK* underpins the two major functional hotspots for $N_2$ fixation: within the euphotic zone and at the top of the anoxic zone (Fig. 2b, h). At Station 2 in the heavily stratified north, the high nitrogenase gene abundances of 70–80 genes per million in the euphotic zone decreased to 10–20 genes per million in underlying oxic waters but experienced a secondary increase to ~250 genes per million at 144 and 156 m within the ACM (Fig. 2b). A similar vertical *nifHDK* distribution was also observed at Station 7 in the upwelling-driven south, with a peak of ~80 genes per million in the euphotic zone, but a distinctly lower secondary peak of ~50 genes per million in the anoxic zone.

Diversity analysis of the recovered *nifH* genes in Lake Tanganyika identified 26 *nifH*-containing metagenome-assembled genomes (MAGs). These MAGs spanned a total of nine putative nitrogen-fixing classes, with the closest relatives indicated in parentheses: Alphaproteobacteria (*Methylocystis*), Acidimicrobiia (IMCC26256), Bdellovibrionota, Brocadiae, Chloroflexia (Roseiflexaceae), Chlorobia (*Chlorobium*), Cyanobacteria (*Dolichospermum*), Gammaproteobacteria (*Aquabacterium*, *Pseudomonas*, *Methylotetracoccus*), Kiritimatiellae (Pontiellaceae), Myxococcota, (Supplementary Fig. 10 and Supplementary Table 2). However, a number of the binned MAGs pertaining to the classes of Alphaproteobacteria, Acidimicrobiia, Bdellovibrionota, Brocadiae, and Chloroflexia did not contain a complete *nifHDK* operon – either as a result of bin incompleteness, or due to the presence of pseudo-*nifH* sequences (solitary *nifH* genes without encoded *nifDK*;[71]). Pseudo-*nifH* sequences, which can comprise a relatively large fraction of curated *nifH* databases, may not be necessarily representative of bonafide diazotrophs as they are unlikely to encode for a functional nitrogenase[71]. The MAGs containing complete *nifHDK* operons were identified within the genera: *Aquabacterium, Chlorobium, Dolichospermum, Methylotetracoccus*, Pontiellaceae (family), and *Pseudomonas* (Fig. 3a, Supplementary Fig. 10, and see Supplementary Tables 3–6 for a complete list of accessory $N_2$ fixation genes). All six *nifHDK*-containing MAGs encoded for the molybdenum-iron (MoFe) nitrogenase, in line with other stratified sulfide-rich lakes[72].

Some diazotrophs were observed at stations in both the north and the south (including Pontiellaceae and *Methylotetracoccus)* (Fig. 2f, l), whereas others were unique to either the north (autotrophs such as *Dolichospermum* and *Chlorobium;* Fig. 2d, f) or south basin (heterotrophs such as *Aquabacterium* and *Pseudomonas*; Fig. 2k, l). Moreover, the diazotroph communities at anoxic depths were distinct from those identified in the oxygenated water column. For example, in the more stratified waters at Station 2, *Dolichospermum* was most abundant in the euphotic zone, whereas *Chlorobium* dominated in the ACM zone, comprising 65% and 85% of the total putative diazotrophic community, respectively (Fig. 2d, f). At Station 7, *Aquabacterium* occurred primarily above the oxic-anoxic transition zone, whereas *Pseudomonas* dominated below. They represented 53% and 61% of the total putative diazotrophic community in each of these zones, respectively (Fig. 2k, l). Altogether, this shows a high degree of both vertical and basin-wide niche partitioning within the diazotrophic community.

This niche partitioning led to contrasts in ecophysiology. While the cyanobacterium *Dolichospermum* carries out oxygenic photosynthesis in well-lit surface waters, *Chlorobium* is an anoxygenic photosynthetic green sulfur bacterium that oxidizes sulfide under extremely low-light conditions, consistent with the light levels in the ACM of Lake Tanganyika (<0.05 μmol photons m$^{-2}$ s$^{-1}$,[47]) and prior observations in deep anoxic waters of many stratified basins and lakes[73,74]. Typical for these microorganisms, *Dolichospermum* and *Chlorobium* MAGs encoded carbon fixation employing the Calvin-Benson-Bassham and the reverse tricarboxylic acid cycles, respectively (Fig. 3b, d and Supplementary Tables 3, 4). Interestingly, they lacked genes for organic matter transport and degradation pathways, suggesting that *Dolichospermum* and *Chlorobium* share a general photolithoautotrophic physiology. To exploit the optimum light intensities and nutrient concentrations within stratified layers, *Dolichospermum* and *Chlorobium* have the genetic potential to control their buoyancy by gas vesicles (Fig. 3b and Supplementary Table 3) for uplift and/or carbohydrate production/consumption for ballasting. Gas vesicle protein genes were not identified in the Lake Tanganyika *Chlorobium*, but *Chlorobium* contained exopolyphosphatases, a polyphosphate kinase, as well as pathways for glycogen synthesis that could moderate buoyancy by metabolizing intracellular polyphosphates and/or glycogen (Fig. 3d and Supplementary Table 4). Competition models have shown that species that control their buoyancy over other non-buoyant microbes have a distinct advantage under more stratified, or less well-mixed, conditions[75]. The putative buoyancy-based ecophysiology of *Dolichospermum* and *Chlorobium* allows them to thrive within the most suitable layer in the heavily stratified water column of the northern basin[47].

In contrast, the upwelling waters of Station 7 were dominated by the gammaproteobacterial diazotrophs *Aquabacterium* and *Pseudomonas*, which exhibited different strategies of carbon cycling and motility (Fig. 2k, l). The recovered *Aquabacterium* and *Pseudomonas* MAGs of Lake Tanganyika included several genes encoding for the uptake and metabolism of high molecular weight compounds (>600 Da), such as those in the dissolved organic matter pool (e.g. TonB-dependent receptors, mono/dioxygenases), as well as genes involved in the uptake and utilization of more labile substrates (e.g., monosaccharides, oligopeptides, and branched-chain amino acids; Supplementary Discussion). The upwelling waters, where *Aquabacterium* and *Pseudomonas* prevail, are characteristically depleted in labile organic matter, with an average DOC:DON ratio of 11.5 that exceeds Redfieldean C:N stoichiometry of 6.6 (Fig. 2i). *Aquabacterium* and *Pseudomonas* can couple the oxidation of organic matter substrates to oxygen and nitrate respiration. To help navigate toward the patchy sources of DOM/POM, the MAGs encoded for flagella-driven chemotaxis. Chemotaxis is known to be advantageous under turbulent conditions, as it allows microbes to sense and follow diffuse gradients of DOM[76]. Therefore, the $N_2$-fixing community and ecophysiology of chemoheterotrophic microorganisms in the upwelling waters contrast with the stratified surface waters and the ACM that are dominated by photoautotrophs.

### Upwelling and stratification influence the mode of diazotrophy

In the more stratified northern basin, high abundances of diazotrophic filamentous cyanobacteria and elevated rates of $N_2$ fixation coincided with low DIN transport to surface waters[44]. Given that P and Fe are not drawn down, the DIN availability in surface waters represents the most likely control over surface $N_2$ fixation by filamentous cyanobacteria in Lake Tanganyika. The northern/central basins provide a favorable environment for communities of photoautotrophic diazotrophs in both the euphotic and anoxic zones (Fig. 4). Concurrently, the capacity for vertical migration (via buoyancy regulation) allows such diazotrophs to navigate precisely in the narrow density gradients of stratified waters.

In contrast, nutrient upwelling established the potential for $N_2$ fixation in the southern basin by adding excess P to surface waters[77]. As a concomitant effect, however, the upwelling of nutrient-rich waters

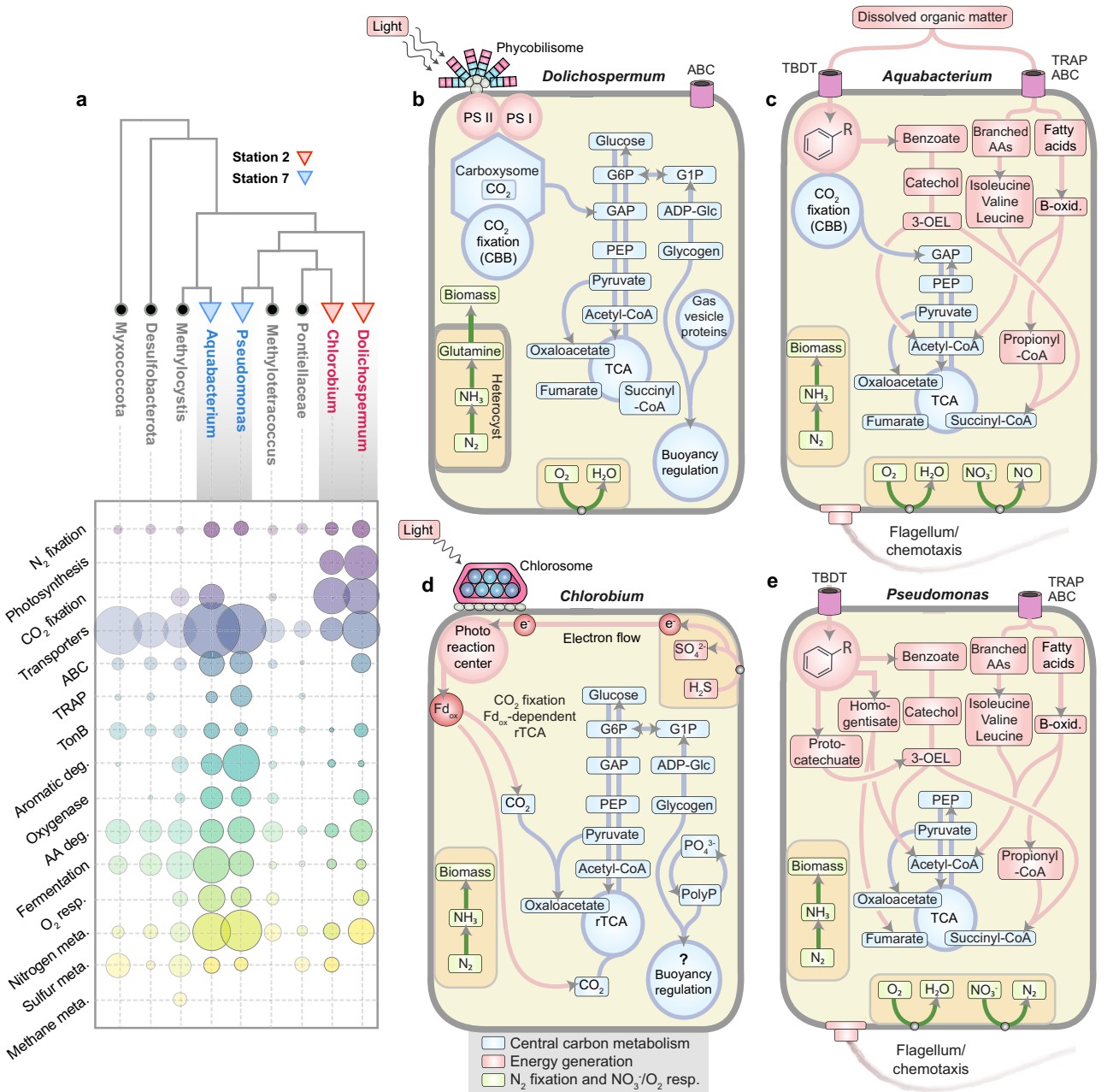

**Fig. 3 | Ecophysiology of key N₂ fixing microbes identified in Lake Tanganyika.** **a** Phylogenetic diversity of recovered *nifH* gene sequences. The unrooted tree was calculated using the maximum likelihood method. Indicated below are the metabolic pathways for the *nifH*-containing diazotrophs. Marker species, specific for Stations 2 and 7 (shown in red and blue), contained the full array of N₂ fixation genes, whereas other members (shown in gray) were missing some key nitrogenase subunits (i.e., contained *nifH*-only), as described in-text. The complete *nifH*, *nifD*, and *nifK* tree is shown in Supplementary Fig. 10. The size of the bubble reflects the total number of genes identified in a given pathway. **b–e** Metabolic models summarizing selected key functions of station-specific marker species. The metabolic models are based on near-complete draft metagenome-assembled genomes with the percent completeness indicated in parentheses: *Dolichospermum* (94.1%), *Chlorobium* (99.4%), *Aquabacterium* (90.4%), and *Pseudomonas* (93.2%) (Supplementary Table 2). For a complete list of genes and a detailed description of metabolic pathways please refer to Supplementary Tables 3–6, and the Supplementary Discussion. G6P glucose-6-phosphate, G1P glucose-1-phosphate, GAP glyceraldehyde 3-phosphate, PEP phosphoenolpyruvate, PolyP polyphosphate, CBB Calvin Bensen-Bassham, rTCA reverse tricarboxylic acid, B-oxid beta oxidation, AA amino acid, 3-OEL 3-Oxodipate-enol-lactone, TBDT TonB-dependent receptors.

seemed to locally suppress key diazotrophs, such as filamentous cyanobacteria, resulting in lower rates of N₂ fixation in surface waters. While filamentous cyanobacteria were inhibited, these conditions favored heterotrophic gammaproteobacteria *Aquabacterium* and *Pseudomonas* which were distinctly reliant on organic matter fluxes. In these productive upwelling waters, their metabolic versatility and chemotaxis make them uniquely suited to a more turbulent and dynamic nutrient landscape, targeting both reactive and semi-reactive

sinking organic material. Chemotactic foraging is favored under dynamic nutrient conditions[78,79], however, it also comes at a high ATP cost. We suggest that ATP allocation towards flagellated motility and metabolic plasticity – used to scavenge for organic nitrogen – could place constraints on the similarly energy-demanding N₂ fixation, explaining the lower rates of N₂ fixation reported in Lake Tanganyika.

Overall, our findings demonstrate that the hydrodynamic regimes – upwelling/mixing versus stratification – represent key controls on

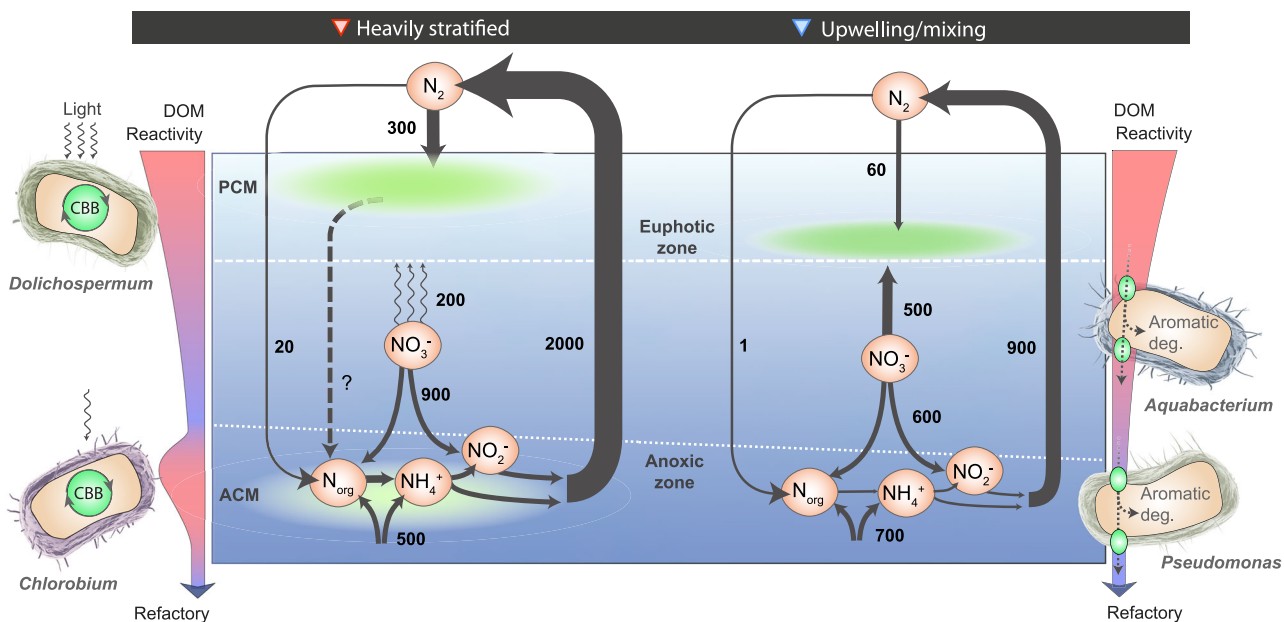

**Fig. 4 | Conceptual model of nitrogen fixation and removal under contrasting hydrodynamic regimes in Lake Tanganyika.** The values indicate the nitrogen fluxes expressed in μmol N m⁻² d⁻¹, see Table 1 for associated errors. Rates of nitrogen removal are detailed in the Supplementary Discussion. In addition, the illustration depicts the key diazotroph marker species identified in the stratified north/center and the upwelling-driven south basins, also indicated by the red and blue downward triangles, respectively. ACM anoxic chlorophyll maximum, PCM primary chlorophyll maximum, DOM dissolved organic matter, CBB Calvin-Benson-Bassham Cycle.

the nutrient landscape that, in turn, governs the mode of diazotrophy (photoautotrophic vs. chemoheterotrophic) and the rates of $N_2$ fixation in Lake Tanganyika.

The observed physical-biological coupling in Lake Tanganyika, a model system with respect to its hydrodynamics, may help to explain why $N_2$ fixation rates are generally low (<5 nM d⁻¹) in some marine upwelling regions[15,22,26–28,30,80,81]. Many of these upwelling systems are also associated with gammaproteobacterial heterotrophic and often chemotactic diazotrophs[79], like *Pseudomonas*[24,31,35,80,82]. Moreover, when highly stratified conditions develop in marine upwelling regions (e.g., coastal sulfidic event[83]), niches for phototrophic green sulfur bacteria become favorable[84], consistent with observations in Lake Tanganyika's Northern basin (Table 1). The predominance of *Dolichospermum* in the stratified North/Central basins of Lake Tanganyika shares some parallels to the stratified ocean gyre systems that support higher abundances of large-sized cyanobacteria (e.g., *Trichodesmium*)[18]. And like *Dolichospermum*, some *Trichodesmium* members are also capable of controlling their buoyancy using gas vacuoles[85], a strategy that possibly contributes to their success as a major $N_2$ fixing species in gyres. Our conceptual hydrodynamic/ecophysiological model presented in Fig. 4, therefore may broadly hold true for other aquatic systems at large.

## Methods

### Physicochemical and biological measurements

Vertical profiles of chemical and biological variables were collected along a lake-wide transect (Fig. 1a) during three cruises at the end of the dry season (28 September - 8 October 2017) and the end of the rainy season (27 April–7 May 2018; 26 April–1 May 2019) onboard *M/V Maman Benita*. The 2018 expedition was the main campaign of this study, combining a high spatial coverage of N process rate measurements, metagenomic analyses of the microbial community, and natural abundance N-isotope tools. During the third campaign in 2019, we collected high-resolution profiles at Station 2 in the north and Station 7 in the south.

Continuous CTD profiles of temperature, photosynthetic active radiation, in-situ chlorophyll fluorescence, turbidity, and oxygen (Sea-Bird SBE 19plus) were followed by discrete water sampling at 5 to 25 m depth intervals using Niskin bottles (20–30 L). Soluble reactive phosphate, ammonium, nitrate, and nitrite concentrations were determined according to standard methods[44], with detection limits averaging 0.22, 0.34, 0.20, and 0.03 μM, respectively. Chlorophyll-*a* concentrations were determined fluorometrically and used to calibrate in-situ chlorophyll fluorescence. The respective samples were extracted in ethanol and calibrated against their respective standards[44]. Medium- to large-celled phytoplankton was concentrated using a 10 μm plankton net, pooling cells from 2–4 L lake water into a 20 mL sample. Samples were fixed with Lugol solution, and cells were counted and classified by inverted microscopy.

### Process rate measurements

Samples for $N_2$ fixation and N turnover incubation experiments were chosen according to the vertical distribution of in-situ chlorophyll fluorescence and oxygen, i.e., targeted at chlorophyll peaks and the top of the anoxic zone. Rates of $N_2$ fixation were determined during incubation experiments at Stations 2 and 9 in September/October 2017 and Stations 2, 4, 6, 7, and 9 in April/May 2018. For determining $N_2$ fixation rates we chose the bubble-removal $^{15\text{-}15}N_2$ tracer method, due to the simple handling and to avoid the risk of trace metal contamination[86]. After sunset, triplicates from each depth were filled from the Niskin into 4.5 L polycarbonate bottles and capped with polypropylene membranes. We used a gas-tight hose to fill the bottles bubble-free. Anoxic samples were filled with an overflow of approximately the bottle volume to minimize contamination with ambient air. The triplicates comprised one control (no added $^{15}N$ label) and two duplicate treatments (with amended $^{15\text{-}15}N_2$). Samples from anoxic depths were collected in 5.3 L glass bottles capped headspace-free with butyl rubber stoppers. To each treatment bottle, 8–10 mL $^{15\text{-}15}N_2$ (Cambridge Isotope Laboratories, Lot # I-21065/AR0664729) were added[87]. The labeled samples were equilibrated for approximately 30 min by rolling/inverting the bottles. Thereafter, the $^{15\text{-}15}N_2$ headspace was released by opening the cap and a 12 mL subsample of water was collected for quantifying the labeling atom percent for each

incubation bottle individually, which averaged 4.7% in 2017 and 6.3% in 2018 sampling campaigns (Supplementary Tables 7 and 8). It should be noted that the bottles were opened only briefly, and without any turbulence, $N_2$ does not equilibrate rapidly ($N_2$ is also inherently much less soluble than oxygen). The remaining air headspace was refilled with water from the same depth. The headspace-free bottles were then transferred to on-deck incubators covered with selected light filters (LEE Filters) to simulate the irradiance and light spectrum at the respective sampling depths. Incubators containing samples from the deep anoxic zone were shaded with black aluminum foil allowing only residual light penetration. Incubations were terminated after 24 h by filtration on pre-combusted GF/F filters (nominal pore size 0.7 μm; Whatman). In our samples, the maximum temperature in the on-deck incubators generally did not exceed the in-situ temperature by more than 4 °C; given that differences were small, rates of $N_2$ fixation were not corrected for temperature. The filters were immediately oven-dried (60 °C for 48 h) and, upon return to land-based facilities, fumed under an HCl atmosphere for 48 h to remove inorganic carbon.

The $^{15}N$ enrichment was measured with an EA-IRMS (vario PYRO cube, Elementar coupled with an IsoPrime IRMS, GV Instruments) and computed according to the following equation (using Ion Vantage Isoprime v1.7.3.0):

$$\delta^{15}N = \left( \frac{15N/14N_{sample}}{15N/14N_{standard}} - 1 \right) 1000 \qquad (1)$$

Calculated limits of detection (LOD = $\delta^{15}N_{contol}$ + 3*standard deviation[$\delta^{15}N_{standards}$]), were unusually low with 3*standard deviation [$\delta^{15}N_{standards}$] <0.4 ‰ in all batches (Supplementary Tables 7, 8). Thus, we conservatively defined the LOD as $\delta^{15}N_{contol}$ + 4 ‰ (minimum change; Montoya et al.[88]). We furthermore defined samples as <LOD, if their $\delta^{15}N$ values or calculated $N_2$ fixation rates were negative (Supplementary Tables 7, 8). Standard error was calculated from all samples of one depth. The $\delta^{15}N$ values of the standards showed no significant linear trend within all analyzed batches (Spearman rank correlation test, $p > 0.05$). PN mass on filters ranged from 12–79 μg N for treatments and 11–62 μg N for controls.

Bulk rates of $N_2$ fixation (in nmol L$^{-1}$ d$^{-1}$) were calculated according to the following equation:

$$N_2\text{fixation rate} = \left( \frac{\text{at\%PON}_{sample} - \text{at\%PON}_{control}}{\text{at\%}N_2 - \text{at\%PON}_{control}} \right) \times \left( \frac{\text{mass of PON per volume}}{\text{time}} \right) \qquad (2)$$

Whereby the at%$^{15}$PON$_{sample}$, at%PON$_{control}$, and at%$N_2$ represent the atomic%$^{15}$N in the particulate organic nitrogen (PON) of the incubated sample, the natural abundance of the control sample, and the $N_2$ pool, respectively. Notably, the control samples (used for natural abundance measurements) were sampled at the same depth as the $^{15-15}N_2$ amendment experiments. The mass of PON per volume (i.e., concentration) and time of incubation were also used as inputs in Eq. 2.

It should be further noted, that trace oxygen contamination in anoxic incubations was unavoidable and may have biased $N_2$ fixation rates in those samples. Assuming that enhanced activity of aerobic diazotrophs in anoxic samples through the introduced amounts of oxygen was the dominant effect, we would expect to measure the highest $N_2$ fixation rates in samples from the anoxic zone in the south, where facultative aerobic diazotrophs, such as *Aquabacterium* or *Pseudomonas*, were more abundant than in the north (Figs. 2 and 3). By contrast, determined $N_2$ fixation rates in anoxic samples were lower in the south than in the north (Figs. 1f and 2a and Supplementary Fig. 5). Moreover, during the collection of the labeling atom percent subsample the introduction of atmospheric $N_2$ may have lowered the actual $^{15-15}N_2$ concentration compared to the measured value. Hence, $N_2$ fixation rate estimates should be considered conservative.

Stable isotope $^{15}$N-labeling experiments were conducted to quantify rates of denitrification and anammox (Supplementary Discussion). For the incubations, waters were collected from Stations 2–9 (Fig. 1), and processed according to Callbeck et al.[47]. A total of four $^{15}$N addition experiments were carried out, at six depths per station: $^{15}$N-NO$_3^-$, $^{15}$N-NO$_2^-$ + $^{14}$N-NH$_4^+$, $^{15}$N-NH$_4^+$ + $^{14}$N-NO$_2^-$, and $^{15}$N-NH$_4^+$. The concentration of $^{14}$N$^{15}$N and $^{15}$N$^{15}$N was then determined in the gas phase by gas-chromatography isotope ratio mass spectrometry (GC-IRMS; VG Optima, Manchester, UK) at land-based facilities. The slope of the linear regression (i.e., $^{14}$N$^{15}$N and $^{15}$N$^{15}$N concentration change with time) was used to calculate the rates of $^{29}$N$_2$ and $^{30}$N$_2$ production in the different experiments.

## POM and nutrient-stable isotopic analyses

Particulate organic matter (POM) samples were collected by filtering 2–4 L lake water onto precombusted GF/F filters (Whatman, UK) and oven-dried at 60 °C for 48 h. At land-based facilities, the POM samples were fumed for 48 h under HCl atmosphere to remove inorganic carbon. The N isotopic composition was analyzed with an EA-IRMS (vario PYRO cube, Elementar coupled with an IsoPrime IRMS, GV Instruments), and calculated according to Eq. 1 (using Ion Vantage Isoprime v1.7.3.0). The final $\delta^{15}$N of the samples was corrected using the standard Acetanilide #1 (Indiana University, CAS # 103-84-4). Standard reproducibility was generally better than 0.5 ‰. Nutrient (DIN) isotope sample treatment and analysis are described in the Supplementary Methods.

## Nitrogen flux estimates

$N_2$ fixation rates were integrated over the top 50 m in the euphotic zone, from 125 to 175 m in anoxic waters, or from 0 to 175 m for total $N_2$ fixation rates. Nitrogen removal rates were integrated down to 175 m. The turbulent diffusive fluxes of nitrate and ammonium were calculated with Fick's law:

$$J_i = K_z(\partial C/\partial z) \qquad (3)$$

The measured concentration gradients; the vertical turbulent diffusivity, $K_z$, was estimated from buoyancy frequency according to Von Rohden et al.[89], and parametrized to a lake-wide average $K_z$ of $10^{-5}$ m$^2$ s$^{-1}$ within the thermocline[41]. The total upward flux of nitrate ($Q$) in the southern upwelling area, i.e., at Stations 8 and 9, was estimated by adding an advective term:

$$Q = J_i + c \times v \qquad (4)$$

Whereby $c$ is the average concentration at the nitracline and $v$ the upwelling velocity (0.05 and 0.1 m d$^{-1}$ at Stations 8 and 9, respectively) estimated from the upward tilting of the isotherms between March and April[42]. Basin-scale estimates were calculated from the average values of Stations 1–6 and 7–9 for the north/central and south basins, respectively. Information regarding the external N sources was extracted from Langenberg et al.[53] by downscaling the annual estimates, while information on the regional variability in atmospheric N deposition was not available. Therefore, the lake-wide estimate was used for both regions.

Nitrate flux values were correlated against various parameters including depth-integrated rates of $N_2$ fixation, $\delta^{15}$N-PON, and cyanobacteria cell densities (Supplementary Fig. 9). Correlative analyses were performed using the Spearman test (one-sided). $R^2$ and $p$-values were evaluated using an 80% and 95% confidence interval.

## Sampling and extraction of DNA and metagenomic analysis

Samples for metagenomic analyses were collected at Stations 2 and 7, similar to our previous study[44]. Briefly, the lake water was filtered onto 0.2 μm cellulose acetate filters, fixed with RNAlater (Sigma Life

Science), and stored at 4 °C for 1–2 weeks. Upon return to land-based facilities, the filters were stored at −80 °C. For extraction, filters were thawed and washed with TE buffer (1x) before DNA extraction using the AllPrep DNA/RNA extraction kit (Qiagen) according to the manufacturer's instructions. The metagenomes were sequenced using the Illumina NextSeq platform to generate 150 bp paired-end reads (averaging ~350 bp in length; Novogene in Hong Kong).

The quality of metagenomic reads was assessed using FastQC (version 0.12.1). Metagenome trimming, quality control, mapping, and taxonomic assignments are detailed in the Supplementary Methods. From the metagenomes, we extracted the genes for $N_2$ fixation (*nifHDK*) from the gtf file. The number of reads for the detected genes was enumerated and normalized for the gene lengths and sequencing depth to obtain a measure of relative gene abundance. The sequencing depth, filtered reads, mean contig length, and mapped reads for each sample are summarized in Supplementary Table 1.

High-quality trimmed reads from all sampling depths ($n = 15$) were co-assembled into scaffolds using Megahit (also see Supplementary Methods). Binning and refinement modules (metaWRAP, version 1.3) were applied to the co-assembly to recover high-quality metagenome-assembled genomes (MAGs). Completeness and contamination rates of the final MAGs were assessed using CheckM (version 1.1.6)[90]. We only used MAGs that passed a threshold for completion of 50 %, and contamination rates less than 10%. MAG abundances were assessed using coverM (version 0.2.0). Here, raw reads were mapped against the putative genomes, and abundance is expressed as the coverage of raw reads on the MAG. The total number of MAGs recovered from our samples, along with bin completeness, and contamination are summarized in Supplementary Table 2.

### Remote sensing

The lake-wide surface chlorophyll distribution was inferred from 300m-resolution raster data from Sentinel-3 OLCI data (Copernicus Global Land Service). The datasets are temporal aggregates of valid observations acquired in the intervals between the first, eleventh, 21st and last day of a month. From version 1.3 of this data set, we extracted the Trophic State Index (TSI) for April and May 2018. The TSI increases in discrete increments of ten that correspond to specific chlorophyll-*a* concentration ranges[91].

### Reporting summary

Further information on research design is available in the Nature Portfolio Reporting Summary linked to this article.

## Data availability

The physicochemical data in this study have been deposited in the ETH Zurich Research Collection: https://www.research-collection.ethz.ch/handle/20.500.11850/418479. The Sentinel-3 OLCI satellite data used in this study can be obtained via the following public database: https://sentinels.copernicus.eu/web/sentinel/missions/sentinel-3. The metagenomic data in this study have been deposited in the NCBI database under the accession code PRJNA675607. Data pertaining to $N_2$ fixation measurements are available in Supplementary Tables 7, 8.

## Code availability

For the analyses used in the manuscript, the software/code is available at the following: Ocean Data View (ODV 5.6.5), https://odv.awi.de; FastQC (v0.12.1), https://github.com/s-andrews/FastQC; Megahit (v1.2.9), https://github.com/voutcn/megahit; metaWRAP (v1.3), https://github.com/bxlab/metaWRAP; CheckM (v1.1.6), https://github.com/Ecogenomics/CheckM; coverM (v0.2.0), https://github.com/wwood/CoverM; Sentinel-3 OLCI data (v1.3), https://land.copernicus.eu/global/.

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

## Acknowledgements

We would like to thank our research collaborators from the Tanzania Fisheries Research Institute: Directors Rashid Tamatamah and Semvua Mzighani as well as Emmanuel A. Sweke, and Mary Kishe. Asante. Many thanks also go to Anthony Kalangali for support in the field and lab. We are very grateful to Mupape Mukuli as well as the captain and crew of the M/V *Maman Benita* for their great help in conducting our research cruises. Special thanks go to Christian Dinkel for his logistic efforts. We also thank Andreas Brand, Julieth B. Mosille, and Tumaini M. Kamulali for field assistance, Damien Bouffard, Alfred Wüest, Martin Schmid, and Tomy Doda for insightful discussions about the hydrodynamics of Lake Tanganyika, Serge Robert, Patrick Kathriner, Jana Tischer, Anja Studer, Philipp Hach, and Gabriele Klockgether for lab assistance, Rosi Siber for GIS analyses, and Eliane Scharmin, Luzia Fuchs, and Patricia Achleitner for their administrative support. We greatly appreciate Wiebke Mohr for the guidance and thoughtful discussions she provided on the manuscript. Many thanks go to our co-collaborators Julian Junker, Ole Seehausen, Catherine E. Wagner, and Tim Kalvelage for launching this project. This study was funded by the Swiss National Science Foundation (CR23I2-166589) and awarded to B. W.

## Author contributions

B.E., B.W., and C.M.C. conceptualized/developed ideas for the study. B.E., C.M.C., K.B.L.B., and A.S.M. collected field samples. B.E., C.M.C., K.B.L.B., R.N., A.S.M., T.K., P.M.M., M.F.L., and D.O.were involved in data analysis. B.W., H.B., I.A.K., M.F.L., and C.J.S. provided resources. B.E. and C.M.C. drafted the original manuscript with substantial input from all co-authors.

## Competing interests

The authors declare that they have no competing interests.
