## [Peer Review File · Nature Communications]

Hydrodynamic regimes modulate nitrogen fixation and the mode of diazotrophy in Lake TanganyikaReviewer #1 (Remarks to the Author):

Ehrenfeld et al. present an interesting, generally well-written and holistic study on the basis of a high-quality dataset on the biogeochemistry of nitrogen fixation in Lake Tanganyika. Using a variety of interdisciplinary methodology in this natural laboratory in which different parts of the same lake experience different hydrodynamic regimes allows them rare, well-constrained insights into the interaction between physical, chemical and microbiological components of an aquatic system.

The main finding is that biological nitrogen fixation (BNF) was considerably higher in the northern, more stratified, part of the lake, compared to the southern part of the lake, which is characterized by upwelling. The authors explain this primarily with limited resupply of dissolved inorganic nitrogen (DIN), and also with enhanced light penetration under stratified conditions. Interestingly, heterotrophic rather than autotrophic diazotrophs predominated under upwelling conditions. They also detected BNF in the anoxic zone of the lake but found that their quantitative contribution to the fixed N pool was insignificant. Furthermore, the authors present detailed information and discussion of the ecophysiology of the main diazotrophs in the lake, which reveal conspicuous vertical and latitudinal patterns.

I have some concerns with the data supporting the main conclusion, which appear somewhat weak, at least as currently presented. While the overall North-South trends (increasing nitrate flux, increasing $\delta^{15}\text{N}$ -PON, decreasing BNF rates; presented in Figure 1) that are underpinning the conclusion seem compelling at first sight, I would urge the authors to discuss the fact that these are correlations that do not automatically imply causation. Moreover, a closer look reveals that some strong claims seem to hinge on a very small number of datapoints. The nitrate flux (Fig 1i) is really quite steady between Stations 1-7 (with a minimum at the shallow central lake area) and the increasing trend only appears in Stations 8 and 9 (Station 7 has slightly lower nitrate flux than Station 1 and 2!). Yet, station 7 is presented (Figure 2) as an example of the unstratified south where BNF is thought to be suppressed by larger nitrate fluxes. (I also note that Station 7 is negative in the thermocline/euphotic boundary parameter depicted in Figure 1h, which should work against plentiful DIN supply to the euphotic zone and thus favor BNF rather than suppress it, see line 154). This is not necessarily saying that I disagree with the interpretation of the data, but more of a longwinded way of illustrating why I think that the data should (also) be presented in a way that more directly allows the assessment of relationships between key parameters underpinning the conclusion. For example, BNF rates are already plotted against cyanobacteria and N-removal in Figure S7. Why not visualize and analyze correlation and/or (multiple) linear regression also for BNF rates vs. nitrate flux and vs the stratification/euphotic zone parameter plotted in Fig. 1h? To give this analysis more analytical power, the authors could also make use of their entire dataset including all three cruises. I think that a more rigorous analysis like this could strengthen the main conclusion - or add nuance in case it suggests that stratification is only one contributor to BNF control. In this context, I noticed that BNF rates seem to also approximately correlate with the trophic-state proxy TSI (Fig 1a). Could trophic state be an additional factor controlling BNF, for example?

I also note that the manuscript is quite long and wordy, which is of course owed to a large part to the enormous variety of data and measurements included. This makes for a holistic overview of BNF in the lake but also for distracting reading at times. In my detailed notes below, I suggested several points at which I found potential to be shorter or more concise and I would like to encourage the authors to do so as much as possible.

Finally, the discussion is relatively short and nevertheless a little wordy, offering few additional insights. I suggest shortening/dissolving (see detailed comments) the current "Discussion" section and adding discussion of bigger-picture aspects instead. Specifically, I would have expected a discussion paragraph about the applicability of the presented findings beyond Lake Tanganyika, maybe extending to the marine realm. The ocean is invoked several times in abstract and introduction and the ambition of the study clearly seems to be to identify controls of BNF that are applicable to aquatic systems more generally. Yet, the relevance of the conclusions to other lakes or the ocean is never really discussed. In fact, the discussion barely looks beyond Lake Tanganyika except for a very short paragraph at the end. While this manuscript holds great potential, I

strongly recommend that a substantive discussion of any wider importance of the findings be added.

Please find detailed comments below.

Detailed comments:

Abstract

Line 29: what explanatory value do the "stable density gradients" add that isn't already covered by 'limited resupply of DIN' and 'enhanced light penetration'?

Introduction

Line 43: the authors might wish to extend this sentence by saying that details of controls are still being worked out. (The following paragraph correctly describes a lot of areas where it is not so clear that e.g. fixed N is always a negative control on BNF.)

Line 45: awkward phrasing – I suggest 'Phosphorus occurs in inorganic (phosphate) and organic form'

Line 48: "...rates even correlate positively..." would sound better to me.

Line 51: please clarify if this sentence refers to lakes or aquatic systems generally.

Line 53: "affected" or "determined" or "controlled" instead of "underpinned"?

Line 92: I think "...which ecophysiological features..." is grammatically correct.

Results:

Line 172: it's not clear to me that this claim is entirely supported by the data, specifically, I don't think we can know that it is the low surface productivity that allowed for the ACM in the north. Figure S1 indicates that the bottom of the oxycline is also deeper in the south compared to the north. The >150m depth of the bottom of the southern oxycline may be too deep for light to sustain anoxygenic photosynthesis (compare Black Sea) even if the south was oligotrophic. So we don't know if it is the differences in surface production or the depth of the bottom of the oxycline that allows an ACM in the north but not in the south. I would guess both. Please rephrase the sentence accordingly.

Line 175: For similar reasons, I am slightly uneasy with this sentence as it seems to assert that the reason for the absence of ACM in the south is due to its high productivity. If the last three words of the sentence ("and vertical mixing") go to my point that the deeper oxycline likely plays a role as well, please make this more explicit.

Line 194: it seems like the only major river flowing into the lake enters at its north end. Is this the reason for the high nitrate concentrations in station 1 (Figure 1 i)? Could riverine DIN input be the reason why BNF and cyanobacteria (Figure 1 l,m) are at their maximum at stations 3 and 4 rather than further north? Might be worth mentioning/discussing.

Line 211: intermediate d15N-PON values at stations 6 and 8 are most closely associated with BNF rates at stations 6 and 7 that are approximately as low as BNF rates at station 9. So the claim that the intermediate d15N-PON values represent intermediate BNF rates doesn't seem entirely correct.

Line 267: Linkage (in the sense of a back and forth between BNF and N-removal) seems not very relevant here given large discrepancy between BNF and N-removal rates (as you write in line 272). Is what you are getting at here simply that BNF can occur in close spatial proximity with N-removal processes which is/was thought to be counterintuitive? If so, and if no actual direct linkage is implied, please consider using "co-occurrence" or similar instead of "linkage" and generally rephrase to express this idea more comprehensively. Also, the rest of this paragraph is about the negligibility of BNF in the AMC so the first sentence seems like a suboptimal (confusing) introduction to this paragraph.

Line 294: this sentence needs rephrasing (grammar).

Line 294-304: this paragraph does not seem very important (with the possible exception of the first two sentences) and could be an opportunity to shorten the manuscript by moving it into the supplementary or removing it completely.

Line 308: "anoxic depths"

Line 341: The discussion in this and the following paragraph contain a lot of details which are not always clearly important for the overall big picture. Consider shortening or moving parts to the supplementary.

Discussion

Line 380: why *molecular* evidence? Many of the rate measurements and key parameters used in this study are not molecular.

Line 383: Consider "favored" or "selected for" instead of "brought out".

Line 387pp: This paragraph and also the one after reads much more like a summary than a discussion; it's long-winded yet seems to add very little of substance that is new at this point. Consider integrating the more important "Discussion" points (especially the discussion about the heterotrophic diazotrophs) into the "Results" section. The current "Results" section is already really a "Results and Discussion" section. (However, I would like to leave it up to the authors how to structure their paper so this is just a suggestion!). In any case, there is certainly potential to make the discussion a lot shorter and more concise (but also add important discussion points that are currently missed, see general comments).

As I write in my general comments, I suggest adding a substantial discussion about how these findings may or may not relate to other lakes or the ocean. This can include the role of stratification, DIN supply and light penetration to BNF, as well as the ecophysiology of the lake's diazotrophs. Are those relevant in the ocean, in other lakes?

Line 390: you talk about "exceptionally high biomass" that may explain high rates of BNF. It's not clear if this refers to a general feature or to Lake Tanganyika specifically. Please clarify, and, if the latter, how does this chime with line 179pp where your BNF rates are described as not particularly high, i.e. somewhere in the mid to lower range compared to other lakes?

Methods

I could not find information on the BNF rate detection limits anywhere. Especially, since they are invoked in the results, please provide them.

Line 506: there should be no comma after "Callbeck et al."

Supplementary Discussion

In the "Effect of N₂ fixation on nitrogen loss" section you write that the correlation between N-removal and N-fixation rates suggest that N-loss may be driven by N-fixation. However, given that N-fixation rates \ll N-removal rates (correct?), wouldn't you expect the causal error in the opposite direction? How can a tiny amount of N-fixation drive large rates of N-loss? Since more N-fixation occurs in the stably stratified regions, isn't it also possible (more likely?) that the stable stratification and resulting anoxia allow for more (anaerobic) N-removal?

Reviewer #2 (Remarks to the Author):

See attached file.

Reviewer #2 Attachment on the following page

SUMMARY

This article presents a comprehensive study of N₂ fixation and diazotroph biogeography in a large model lake. The authors analyzed standard hydrographic parameters including nutrients, N₂ fixation rates, denitrification rates, and metagenomes. The major findings are as follows:

1. Upwelling affects the distribution of surface N₂ fixation and diazotrophic filamentous cyanobacteria.
2. Surface N₂ fixation in the northern and central basin, where resupply of dissolved inorganic N via upwelling is low, likely fuels new production.
3. N₂ fixation rates within deep anoxic waters were “low but persistent” [line 246].
4. Diazotrophs displayed clear biogeographic patterns, with chemotactic heterotrophic (putative) diazotrophs dominating in upwelling waters.

RESEARCH CONTEXT AND REVIEWER PERSPECTIVE

Understanding the role of physico-chemical dynamics in regulating aquatic N₂ is essential to closing N budgets, on both local and global scales, and predicting future change. Controls on N₂ fixation remain poorly constrained, largely due to the metabolic diversity of extant diazotrophs, and represent a major area of uncertainty in our understanding of the global N cycle and, due to the importance of N limitation to aquatic productivity, the carbon cycle as well.

This study offers insight into N cycling in a model lake—an aquatic system in which N dynamics are currently under-studied. While the role of the major hydrodynamic phenomenon discussed here, upwelling, in regulating N₂ fixation has been the center of a lot of attention in marine systems, the diazotrophs which this article reports from Lake Tanganyika are uncommon in most marine upwelling systems. Consequently, I believe this article provides an interesting perspective and will be a useful addition to the body of literature on the topic.

On the whole, I found the article to be articulate and the study to be comprehensive. Below, I list some specific comments which I believe will enhance the article if addressed. Excepting only the first of my major comments, I do not expect any changes which I suggest to alter the major findings of the article.

MAJOR COMMENTS

- The authors do not discuss how N₂ fixation rate detection limits were assessed (or how rates and rate error was calculated). This is a major omission that has the potential to change major point #3 listed above. How to assess the detectability of N₂ fixation has been a major issue within the community in the past five years (see White et al. 2020, cited in the MS). “Low but persistent” rates have been increasingly reported from deep waters; however, in many cases these may represent methodological artifacts (see Selden et al. 2021 doi: 10.1002/Ino.11735). Low rates, when averaged across the large volume of the world’s deep waters, could significantly alter the global N budget (see Benavides et al 2018 doi: 10.3389/fmars.2018.00108 for discussion of the potential significance), so please consider them carefully.
- Similarly, to facilitate downstream use of the N₂ fixation rate data, the authors should report all values used to calculate rates (and all rates in tabular format)—either in a data repository or in a supplemental table. Best data practices are discussed in White et al 2020.
- I recommend that the authors revisit the references in the introduction. There are a few points where the inclusion of key references may change the authors’ statements. See my comments below.

MINOR COMMENTS

- [line 43] These seemed like an odd assortment of references to choose, and why are they only listed for P? The potential for Fe limitation of N₂ fixation was first proposed by Falkowski. Fixed N as a control is reviewed by Knapp 2012, who cites original literature on the topic (from the late '90s, early '00s).
- [47] DIN species as a control differs by diazotroph. I recommend citing papers from the genetics literature where, for common taxa, regulatory pathways are well-studied (eg Dixon & Kahn 2004, Nat Rev Microbiol).
- [60-61] Regarding upwelling references and the "5 nmol L⁻¹ d⁻¹" value – These papers are peculiar choices for several reasons. First, there may have been methodological issues with the cited studies (discussed in Selden et al 2021 doi: 10.1002/lno.11735). Second, there have been numerous studies from other coastal upwelling systems like the eastern tropical north pacific where rates can be on the order of 10-100 nmol N L⁻¹d⁻¹ (eg Turk-Kubo et al 2021 doi: s43705-021-00039-7; Selden et al 2019 doi: 10.1029/2019GB006242).
- [64] Decoupling due to Fe limitation is likely more important (Weber & Deutsch 2014 www.pnas.org/cgi/doi/10.1073/pnas.1317193111).
- [66] What do the authors consider to be "hotspots"? To my mind, hotspots are where rates are exceptionally high. 5 nmol N L⁻¹ d⁻¹ is not exceptionally high compared to coastal waters where even low N₂ turnover comes to be a high rate due to high biomass (eg in Mid-Atlantic Bight -- Mulholland et al. 2019 GBC; Selden et al 2021 doi: 10.1002/lno.11727; Tang et al 2020 [10.1038/s41396-020-0703-6](https://doi.org/10.1038/s41396-020-0703-6)).
- [71] 5 nmol N L⁻¹ d⁻¹ is not that high compared to N₂ fixation rates in high biomass coastal waters. See refs above, among others.
- [82] The ecophysiology of N₂ fixation by non-cyanobacterial autotrophic diazotrophs is not well understood. Consider rephrasing.
- [145] I recommend adding a sentence here noting that nitrate was almost completely drawn down below ~150 m (due to denitrification or anammox?)
- [184] Ref 29 is inappropriate here—not an upwelling study
- [198-212] Can the authors comment as to whether denitrification/anammox may have affected del15N observed in upper waters, particularly where denitrified waters are upwelled?
- [242] These are not recent ideas
- [246] Per my comment above, please consider detection and quantification limits very carefully before asserting that rates are "low but persistent" in deep waters. Unless [PN] is very very low, I am dubious that <1 nmol N⁻¹ d⁻¹ can be distinguished from noise with high confidence.
- [272] Can surface N₂ fixation make up for fixed N loss below?
- [282] nifH alone does not encode a functioning nitrogenase. Please make note of this somewhere so that readers can understand why the difference between nifH-encoding metagenomes and nifHDK-encoding metagenomes is important.
- Did nifHDK-encoding MAGs also have the genes for the biosynthetic pathway ie nifENB?
- [296-303] Fe-only and V-encoding nitrogenases almost always co-occur with Mo-nitrogenase (Poudel et al. 2018 <https://doi.org/10.1128/JB.00757-17>). Is it possible that both were present?
- [352] Is the C:N of the diazotrophs found here known?
- [469] How were anoxic N₂ fixation incubations sampled i.e., did the authors use standard techniques for sampling low-O₂ waters (overflowing 3x, no bubbles, etc)?
- [477] Are the "membranes" used here different from the standard serum caps used with PETG incubation bottles?
- [481] How did the authors establish that stocks were ammonium-free?

- [483] Why not use a syringe to remove the bubble? Did the authors open even anoxic incubations? Can the authors comment on whether the introduction of new air may have affected the dissolved [N₂] concentration i.e., could dissolved ¹⁵N have decreased as it equilibrated with this new light N air?
- [483] What do the authors mean by “shaking”? Most people gently roll incubation vials to avoid rupturing cells.
- [489] What was this average value? Please report all data used to calculate rates for all incubations in a table somewhere, following best data practices in N₂ fixation reporting (see White et al 2020).
- [503] Was PN at t=0 collected? How were detection limits calculated? How was error calculated? Was AN₂ measured for each incubation (apart from in 2019)? What was the range of those values? What was the PN mass on the filters used to get PN atom% values? What were the linearity criteria used in this study? How were those criteria established? There is a lot of missing information here. It could all go in supplemental text, but needs to be included. Please revisit White et al 2020.
- [551] How long did the filters sit before freezing? Was this within limits recommended by the manufacturer?

We greatly appreciate the two reviewers for their support of this work, their attention to detail and their constructive criticism. Below is a point-by-point response to the reviewers' comments, indicated in the red-type face. The grey highlighted line numbers refer to changes made in the revised manuscript (e.g lines XX-XX). A complete list of references is included at the end of the response to reviewers' comments.

Reviewer #1 (Remarks to the Author):

Ehrenfeld et al. present an interesting, generally well-written and holistic study on the basis of a high-quality dataset on the biogeochemistry of nitrogen fixation in Lake Tanganyika. Using a variety of interdisciplinary methodology in this natural laboratory in which different parts of the same lake experience different hydrodynamic regimes allows them rare, well-constrained insights into the interaction between physical, chemical and microbiological components of an aquatic system.

The main finding is that biological nitrogen fixation (BNF) was considerably higher in the northern, more stratified, part of the lake, compared to the southern part of the lake, which is characterized by upwelling. The authors explain this primarily with limited resupply of dissolved inorganic nitrogen (DIN), and also with enhanced light penetration under stratified conditions. Interestingly, heterotrophic rather than autotrophic diazotrophs predominated under upwelling conditions. They also detected BNF in the anoxic zone of the lake but found that their quantitative contribution to the fixed N pool was insignificant. Furthermore, the authors present detailed information and discussion of the ecophysiology of the main diazotrophs in the lake, which reveal conspicuous vertical and latitudinal patterns.

I have some concerns with the data supporting the main conclusion, which appear somewhat weak, at least as currently presented. While the overall North-South trends (increasing nitrate flux, increasing d15N-PON, decreasing BNF rates; presented in Figure 1) that are underpinning the conclusion seem compelling at first sight, I would urge the authors to discuss the fact that these are correlations that do not automatically imply causation. Moreover, a closer look reveals that some strong claims seem to hinge on a very small number of datapoints. The nitrate flux (Fig 1i) is really quite steady between Stations 1-7 (with a minimum at the shallow central lake area) and the increasing trend only appears in Stations 8 and 9 (Station 7 has slightly lower nitrate flux than Station 1 and 2!). Yet, station 7 is presented (Figure 2) as an example of the unstratified south where BNF is thought to be suppressed by larger nitrate fluxes. (I also note that Station 7 is negative in the thermocline/euphotic boundary parameter depicted in Figure 1h, which should work against plentiful DIN supply to the euphotic zone and thus favor BNF rather than suppress it, see line 154). This is not necessarily saying that I disagree with the interpretation of the data, but more of a longwinded way of illustrating why I think that the data should (also) be presented in a way that more directly allows the assessment of relationships between key parameters underpinning the conclusion. For example, BNF rates are already plotted against cyanobacteria and N-removal in Figure S7. Why not visualize and analyze correlation and/or (multiple) linear regression also for BNF rates vs. nitrate flux and vs the stratification/euphotic zone parameter plotted in Fig. 1h? To give this analysis more analytical power, the authors could also make use of their entire dataset including all three cruises. I think that a more rigorous analysis like this could strengthen the main conclusion - or add nuance in case it suggests that stratification is only one contributor to BNF control. In this context, I noticed that BNF rates seem to also approximately correlate with the trophic-state proxy TSI (Fig 1a). Could trophic state be an additional factor controlling BNF, for example?

We thank the reviewer for the substantive discussion regarding visual trends and the suggestion to add correlative analyses to better emphasize the conclusions of the work. The reviewer states that the "nitrate flux is quite steady between stations 1-7", and in this regard, we admit that this trend appears the case, as stations 8 and 9 exhibit much higher nitrate fluxes that, in turn, skew the axis range (Fig. 1i). However, if one were to zoom into the nitrate flux data across the lake (see below), then clear differences emerge. In the figure, the nitrate flux for stations 1-7 can range from

0.15 to 0.3 mmol m⁻² d⁻¹ (a 2-fold difference) and is thereby substantial, but overshadowed by the larger differences observed at stations 8 and 9. If we overlay the filamentous cyanobacteria abundances and N₂ fixation rates on top of the nitrate flux data (see below), then cyanobacteria and their associated N₂ fixation are clearly occurring in the North/Center of the lake where the nitrate flux is lowest. Whereas, the DIN input from fluvial sources (as discussed in our comment below) at the northernmost station and upwelling in the south hampered cyanobacteria and their associated N₂ fixation at the extremities of the lake. Hence, we are confident that the pattern viewed in the figure below provides a fairly compelling link.

Figure: Distribution of filamentous nitrogen-fixing cyanobacteria, rates of nitrogen fixation and the nitrate flux across a north-south transect in Lake Tanganyika. The transect and profiles shown were taken in April/May 2018.

However, the reviewer’s point is well taken that correlative analyses could provide a stronger argument for the proposed mechanism (as sketched e.g., in Fig. 4). Across the lake, the abundance of filamentous cyanobacteria exhibited a significant negative correlation with the nitrate flux ($R^2 = 0.79$, $p < 0.01$; $N = 9$; revised Fig. S9a, or figure below). Moreover, the rates of N₂ fixation showed a trend towards higher activity with a decreasing nitrate flux, albeit this trend was not statistically significant due to the limited number of stations with available N₂ fixation data ($R^2 = 0.50$, $p > 0.05$; $N = 5$; Fig. S9d). In addition to a limited number of stations, N₂ fixation rates may suffer from a lagged response by the diazotroph community towards environmental changes, for instance, the nitrate supply (Ehrenfels et al. 2021). Nevertheless, the correlation of **1**) the nitrate flux with the abundances of filamentous cyanobacteria (Fig. S9a), and **2**) filamentous cyanobacteria abundances with N₂ fixation activity ($R^2 = 0.97$; $p < 0.01$; $N = 5$; Fig. S9b), together affirm that DIN resupply is a key modulating factor of filamentous cyanobacteria and associated N₂ fixation in Lake Tanganyika. Therefore, we attribute the high N₂ fixation rates by filamentous cyanobacteria in the north and center to the low resupply of DIN, which provides the basis for a competitive advantage to N₂-fixing phytoplankton. We have added the above information on lines 203-212 in the revised manuscript.

We have included other correlative analyses, such as the distance between the euphotic depth (Z_{eu}) and the thermocline ($Z_{eu} - TC$) versus the abundances of filamentous cyanobacteria, N₂ fixation rates and the nitrate flux. The $Z_{eu} - TC$ was significantly correlated with the abundances of filamentous cyanobacteria (Fig. S9c), but not significantly correlated with the nitrate flux and N₂ fixation rates (Fig. S9e, f). Thus, the $Z_{eu} - TC$ may serve as another indicator/predictor of where niches may, or may not, exist for filamentous cyanobacteria in Lake Tanganyika. The reviewer is also correct to point out that TSI appears to link spatially with N₂ fixation rates and the distribution of filamentous cyanobacteria. Given the pigmented nature of cyanobacteria it indeed makes sense that we observe higher chlorophyll a in the North/Central basins where filamentous cyanobacteria co-occur. At the moment, we have not made any further correlations of TSI with N₂ fixation or filamentous cyanobacteria abundances, as the other parameters mentioned above provide sufficient context for the conclusions highlighted in Fig. 4. Again, we appreciate the suggestion by the reviewer to include correlative analyses, which have provided more support for the conclusions in the manuscript.

The reviewer also suggested performing this analysis together with the data collected from our previous campaigns. At present we have limited our statistical analyses to the 2018 campaign (seen in Fig. S9), and have not included data from 2017 (Sep/Oct, N = 2) and 2019 (Apr/May; N = 2) mainly because they are not complete transects spanning from North to South, and because seasonal/interannual differences have been observed in Lake Tanganyika. If we had data spanning multiple years over different seasonal time points, then a larger statistical analysis that combines all data/sampling campaigns, and that provides additional and more robust constraints on seasonal fluctuations, may be worthwhile. However, the 2018 dataset remains the most comprehensive and conclusive N₂ fixation transect across the lake thus far.

New Fig. S9: Correlation matrix with the rates of N₂ fixation, the abundance of filamentous, diazotrophic cyanobacteria, the distance between the euphotic depth (Z_{eu}) and the thermocline ($Z_{\text{eu}} - \text{TC}$), as well as the estimated nitrate flux (NO_3^- flux). A linear regression line is shown with R^2 and p -values (Spearman rank correlation test) indicated above each panel.

I also note that the manuscript is quite long and wordy, which is of course owed to a large part to the enormous variety of data and measurements included. This makes for a holistic overview of BNF in the lake but also for distracting reading at times. In my detailed notes below, I suggested several points at which I found potential to be shorter or more concise and I would like to encourage the authors to do so as much as possible.

We greatly appreciate the reviewer's suggestions regarding manuscript readability and length. The reviewer is correct in pointing out that some sections did in fact require some careful trimming in order to remove unnecessary details, or to move information where possible to the supplementary discussion. The largest trimming occurred in the introduction (lines 35-88), in the metagenomics section (lines 263-338), and in the last discussion section (lines 339-377) of the

revised manuscript. Overall, we have reduced the word count by nearly 800-words (or 3 pages) compared to the original manuscript, thereby improving the readability of the manuscript.

Finally, the discussion is relatively short and nevertheless a little wordy, offering few additional insights. I suggest shortening/dissolving (see detailed comments) the current “Discussion” section and adding discussion of bigger-picture aspects instead. Specifically, I would have expected a discussion paragraph about the applicability of the presented findings beyond Lake Tanganyika, maybe extending to the marine realm. The ocean is invoked several times in abstract and introduction and the ambition of the study clearly seems to be to identify controls of BNF that are applicable to aquatic systems more generally. Yet, the relevance of the conclusions to other lakes or the ocean is never really discussed. In fact, the discussion barely looks beyond Lake Tanganyika except for a very short paragraph at the end. While this manuscript holds great potential, I strongly recommend that a substantive discussion of any wider importance of the findings be added.

We agree with the reviewer that our connection back to marine systems was lacking in the discussion. To remedy this shortfall, we have dedicated a section to addressing the potential implications of our findings for the marine realm. The new section can be found on lines 364-377 of the revised manuscript. Therein, we take the reader across a transect of the ocean, spanning from near-coastal upwelling (sulfide-rich) to the offshore upwelling, and lastly, into the open ocean gyre. This transect not only traverses clear differences in DIN supply and hydrodynamics, but also highlights the changes in the marine diazotrophic communities. The crux of this section argues that our findings in Lake Tanganyika (while different with respect to the exact hydro-biogeochemistry) share some clear resemblances with ocean systems.

We would also like to point out that we have subsequently replaced the original “future outlook” section with a paragraph describing our works implications for the marine realm. Given that both sections are more speculative in nature, we believe this was a fair tradeoff. Moreover, this helps to maintain a low/similar word count, while also drawing connections to other environments.

Please find detailed comments below.

Detailed comments:

Abstract

Line 29: what explanatory value do the “stable density gradients” add that isn’t already covered by ‘limited resupply of DIN’ and ‘enhanced light penetration’?

We agree with the reviewer that “stable density gradients” was superfluous. It was dropped in the revised manuscript on line 29.

Introduction

Line 43: the authors might wish to extend this sentence by saying that details of controls are still being worked out. (The following paragraph correctly describes a lot of areas where it is not so clear that e.g. fixed N is always a negative control on BNF.)

We thank the reviewer for highlighting this point. Indeed, the statement was overly broad, and we have therefore qualified this statement by providing additional nuance, it now reads as (lines 38-39), “Nitrogen fixation itself may be regulated by the availability of phosphorus (P), iron (Fe), and/or fixed N (Falkowski 1997; Mills et al. 2004; Knapp 2012; Wasmund et al. 2012; Romero et al. 2013; Schoffelen et al. 2019)”

Line 45: awkward phrasing – I suggest ‘Phosphorus occurs in inorganic (phosphate) and organic form’

We agree with the reviewer, but have since dropped this line as phosphate in either its inorganic or organic form was not critical information. In the revised manuscript, the statement reads as follows (line 39), “While phosphorus is an essential macronutrient for the growth of all organisms...”

Line 48: "...rates even correlate positively..." would sound better to me.

We agree with the reviewer's suggestion and have therefore made this change in the revised manuscript (line 42).

Line 51: please clarify if this sentence refers to lakes or aquatic systems generally.

We have since revised this part of the introduction, and shortened the introduction overall.

Line 53: "affected" or "determined" or "controlled" instead of "underpinned"?

As above, we have revised this section of the introduction, and shortened the introduction overall.

Line 92: I think "...which ecophysiological features..." is grammatically correct.

See above.

Results:

Line 172: it's not clear to me that this claim is entirely supported by the data, specifically, I don't think we can know that it is the low surface productivity that allowed for the ACM in the north. Figure S1 indicates that the bottom of the oxycline is also deeper in the south compared to the north. The >150m depth of the bottom of the southern oxycline may be too deep for light to sustain anoxygenic photosynthesis (compare Black Sea) even if the south was oligotrophic. So we don't know if it is the differences in surface production or the depth of the bottom of the oxycline that allows an ACM in the north but not in the south. I would guess both. Please rephrase the sentence accordingly.

We thank the reviewer for raising this interesting discussion. Indeed, in complex environments, it is often a multitude of factors that contribute to determining a particular biogeochemical feature. We agree with the reviewer that the positioning of the oxycline plays a role in determining the presence or absence of the ACM, and we have clarified this point on lines 140-142, where it now reads as, "This, combined with the fact that, vertical mixing causes a deepening of the oxycline could additionally constrain the available niche for anoxygenic phototrophic sulfur bacteria."

At the same time, we still subscribe to the idea that surface primary productivity plays a role constraining light penetration to deeper depths. Briefly, by all available data (CTD-based Chl *a* fluorescence, pigment analysis, CO₂ fixation rates, and historical trends) we have generally lower and higher primary productivity in the North and South basins, respectively (Verburg and Hecky 2003; O'Reilly et al. 2003; Bergamino et al. 2010; Callbeck et al. 2021b). In the Northern basin, the reduced upwelling of nutrients diminishes epilimnetic primary productivity and increases the water clarity, as reported elsewhere (Verburg 2003; O'Reilly 2003). While both stations exhibit strong light attenuation with depth, light penetrated 20% deeper in the North (down to 100 m) compared to the South (down to 80 m), respectively (please find the light irradiance profiles below, which are now included as Fig. S2 in the revised manuscript). Hence, the deeper light penetration, combined with the relatively shallow oxycline compared to the south, provide green sulfur bacteria with improved access to light in the northern basin. And once established in the North, we believe these anoxic phototrophic blooms contributed to a greater degree of oxycline shoaling, in part, because the ACM was associated with enhanced rates of organic matter respiration (see Callbeck et al., 2021 for further details). Hence, we acknowledge on lines 140-142 that both light penetration (driven by primary productivity) and vertical mixing are potential factors constraining the available niche for anoxygenic phototrophic sulfur bacteria.

New Fig. S2. Vertical light irradiance profiles for Lake Tanganyika. Data shown is from the April-May 2019 campaign. (a, b) The photosynthetically active radiation is expressed as the photon flux density (PFD) for stations 2 and 7, which are indicated in the top and bottom panels, respectively. The corresponding surface light irradiance values, and the time of day when the measurements were performed are indicated in the adjacent table.

Line 175: For similar reasons, I am slightly uneasy with this sentence as it seems to assert that the reason for the absence of ACM in the south is due to its high productivity. If the last three words of the sentence (“and vertical mixing”) go to my point that the deeper oxycline likely plays a role as well, please make this more explicit.

We agree with reviewer (also see above comment), and have therefore added to the revised manuscript on lines 140-142: "This, combined with the fact that, vertical mixing causes a deepening of the oxycline could additionally constrain the available niche for anoxygenic phototrophic sulfur bacteria."

Line 194: it seems like the only major river flowing into the lake enters at its north end. Is this the reason for the high nitrate concentrations in station 1 (Figure 1 i)? Could riverine DIN input be the reason why BNF and cyanobacteria (Figure 1 l,m) are at their maximum at stations 3 and 4 rather than further north? Might be worth mentioning/discussing.

The reviewer raises an interesting point. Indeed, two major river inlets are situated in the north of Lake Tanganyika. Hence, the higher nitrate concentrations at station 1 may be attributable to the river flux. As the reviewer correctly points out, the fluvial input of DIN (Fig. 1c), likely negated the competitive advantage of N_2 -fixing phytoplankton in this particular region (Fig. 1g, m). In contrast, at stations 3 and 4 located further south, which display high diazotrophic cyanobacteria abundances and rates of N fixation, our previous work has found no evidence of riverine signals (discussed in detail in Ehrenfels et al., 2021). Instead of adding this information here, we decided to append this to our broader discussion a few paragraphs below on lines 215-222 of the revised manuscript where it reads as follows:

“An exception to this trend is the northernmost Station 1, which was largely devoid of N₂-fixing cyanobacteria (Fig. 1g, m). Here, the relatively high availability of DIN in surface waters (Fig. 1c, i), which may have been injected from deeper layers by internal waves (Naithani and Deleersnijder 2004), or may stem from riverine sources (Langenberg et al. 2003), likely negated the competitive advantage of N₂-fixing phytoplankton as discussed elsewhere (Ehrenfels et al. 2021). Likewise, in the south basin, the wind-driven upwelling of nutrient-rich waters impeded the proliferation of filamentous cyanobacteria. Instead, these waters were dominated by heterotrophic diazotrophs (discussed below), and were associated with much lower N₂ fixation rates compared to the north basin.”

Line 211: intermediate d15N-PON values at stations 6 and 8 are most closely associated with BNF rates at stations 6 and 7 that are approximately as low as BNF rates at station 9. So the claim that the intermediate d15N-PON values represent intermediate BNF rates doesn't seem entirely correct. We agree with the reviewer that a gradual transition was not clearly evidenced by the data, hence the language in this statement was rather imprecise. Instead, we see a clear contrast between the North and South basins both with respect to N₂ fixation rates and the ¹⁵N-PON signatures. We have subsequently dropped the original wording that referred to intermediate values/stations and replaced it with (lines 180-182), “corroborating the north-south differences in surface N₂ fixation across the lake”. We hope this provides additional clarity in the revised manuscript.

Line 267: Linkage (in the sense of a back and forth between BNF and N-removal) seems not very relevant here given large discrepancy between BNF and N-removal rates (as you write in line 272). Is what you are getting at here simply that BNF can occur in close spatial proximity with N-removal processes which is/was thought to be counterintuitive? If so, and if no actual direct linkage is implied, please consider using “co-occurrence” or similar instead of “linkage” and generally rephrase to express this idea more comprehensively. Also, the rest of this paragraph is about the negligibility of BNF in the AMC so the first sentence seems like a suboptimal (confusing) introduction to this paragraph.

We agree with the reviewer that the opening sentence to this paragraph was misleading. We have subsequently removed the word “high” and replaced it with “measurable” rates. In addition, we have now made clear that while a co-occurrence of N fixation and N loss may exist, the extent to which newly fixed N by biological N fixation sustains N loss is, however, negligible. In addition, we have moved a sentence describing the rates of N loss to the supplement, shortening this paragraph and its main message. We hope that this has provided additional clarity, please see lines 250-256 of the revised manuscript where it now reads as:

“The measurable rates of N₂ fixation in the anoxic zone, raised the possibility of a co-occurrence between N₂ fixation and N removal, which has been previously observed in other stratified lakes and basins (Halm et al. 2009; Farnelid et al. 2013). While such a co-occurrence may exist, rates of N₂ fixation in the anoxic zone satisfied at most 1 % of the fixed N required to balance N loss (Table 1). Instead, the export of newly-fixed N from filamentous cyanobacteria in surface waters could contribute up to 12% of the fixed N lost in the northern oxygen-deficient zone (Table 1).”

Line 294: this sentence needs rephrasing (grammar).

In the revised manuscript (lines 283-286), it now reads as, “The MAGs containing complete *nifHDK* operons were identified within the genera: *Aquabacterium*, *Chlorobium*, *Dolichospermum*, *Methylotetracoccus*, *Pontiellaceae* (family), and *Pseudomonas* (Fig. 3a; Fig. S10, see Table S3-S6 for a complete list of accessory N₂ fixation genes).”

Line 294-304: this paragraph does not seem very important (with the possible exception of the first two sentences) and could be an opportunity to shorten the manuscript by moving it into the supplementary or removing it completely.

We thank the reviewer for raising this point and highlighting areas of the manuscript that could be shortened. We have rephrased this sentence for clarity and have subsequently dropped the three sentences that followed as they were superfluous to the discussion – please find the revised sentence on lines 286-288.

Line 308: “**an**oxic depths”

We have now changed “in” to “at”, in the revised manuscript (line 240).

Line 341: The discussion in this and the following paragraph contain a lot of details which are not always clearly important for the overall big picture. Consider shortening or moving parts to the supplementary.

We thank the reviewer for the suggestion; we also agree that some of these details distracted from the main message. We have therefore condensed three paragraphs down to only two (see lines 301-338), with the word count decreasing by 50%, with some of this information now moved to the supplementary discussion under the section title “*Dolichospermum* and *Chlorobium*” and “*Aquabacterium* and *Pseudomonas*”. At the same time, we were careful not to remove too much detail, as heterotrophic diazotrophs are poorly described in the literature, especially with respect to the type of heterotrophy they can perform (heavy vs. labile organic matter degraders). Here, we would like to emphasize the particular pathways that define these diazotrophs and their strategy for thriving in these waters, see Fig. 3c, e). Some detail is therefore necessary, but we have trimmed this as much as possible.

Discussion

Line 380: why *molecular* evidence? Many of the rate measurements and key parameters used in this study are not molecular.

In this particular case we were referring to the abundance of *nif* genes in the metagenomes presented in Fig. 2c-e, and panels h-j. Nevertheless, we have dropped this summary paragraph as we have changed the manuscript from separate results and discussion to a combined results/discussion style (also according to a comment by Reviewer 2), and hence, the summary paragraph was unnecessary.

Line 383: Consider “favored” or “selected for” instead of “brought out”.

We have since removed our summary paragraph in order to streamline the manuscript, also according to our response above.

Line 387pp: This paragraph and also the one after reads much more like a summary than a discussion; it’s long-winded yet seems to add very little of substance that is new at this point. Consider integrating the more important “Discussion” points (especially the discussion about the heterotrophic diazotrophs) into the “Results” section. The current “Results” section is already really a “Results and Discussion” section. (However, I would like to leave it up to the authors how to structure their paper so this is just a suggestion!). In any case, there is certainly potential to make the discussion a lot shorter and more concise (but also add important discussion points that are currently missed, see general comments).

We agree with the reviewer that there is potential to trim some areas of this section. We have therefore removed the first summary paragraph that preceded this section, as it was superfluous to the manuscript. In addition, we have re-organized the separate “Results” and “Discussion” sections, into the “Results and Discussion” section as this proved more suitable for discussing the complex data presented. Moreover, we have moved information regarding the rates of nitrogen fixation and its positive correlation with rates of nitrogen loss to the supplemental discussion. Overall, we have reduced the size of this section by nearly one-half, while maintaining the take home message and while implementing a small paragraph discussing the implications of our findings for the marine realm (please see the revised discussion on lines 339-377).

As I write in my general comments, I suggest adding a substantial discussion about how these findings may or may not relate to other lakes or the ocean. This can include the role of stratification, DIN supply and light penetration to BNF, as well as the ecophysiology of the lake’s diazotrophs. Are those relevant in the ocean, in other lakes?

We agree with the reviewer that our connection back to marine systems was lacking in the discussion, please see also the response in the general comments above. To remedy this shortfall, we have dedicated a paragraph to addressing the potential implications of our findings to the marine realm. The new paragraph is found on lines 364-377 where it reads as follows:

“The observed physical-biological coupling in Lake Tanganyika, a model system with respect to its hydrodynamics, may help to explain why N₂ fixation rates are generally low (< 5 nM d⁻¹) in marine upwelling regions (Luo et al. 2014; Knapp et al. 2016; Bonnet et al. 2017; Wang et al. 2019; Selden et al. 2021; Kittu et al. 2023); and why upwelling systems are

associated with gammaproteobacterial heterotrophic and often chemotactic diazotrophs (Hallström et al. 2022), like *Pseudomonas* (Jayakumar and Ward 2020). Moreover, when highly stratified conditions develop in marine upwelling regions (e.g., coastal sulfidic event (Callbeck et al. 2021a)), niches for phototrophic green sulfur bacteria become favorable (Ma et al. 2021), consistent with observations in Lake Tanganyika's Northern basin (Table 1). The predominance of *Dolichospermum* in the stratified North/Central basins of Lake Tanganyika shares some parallels to the stratified ocean gyre systems that support higher abundances of large sized cyanobacteria (e.g., *Trichodesmium*) (Pierella Karlusich et al. 2021). And like *Dolichospermum*, some *Trichodesmium* members are also capable of controlling their buoyancy using gas vacuoles (Villareal and Carpenter 2003), a strategy that possibly contributes to their success as a major N₂ fixing species in gyres. Our conceptual hydrodynamic/ecophysiological model presented in Fig. 4, therefore may broadly hold true for marine systems, and other aquatic systems at large.”

Line 390: you talk about “exceptionally high biomass” that may explain high rates of BNF. It's not clear if this refers to a general feature or to Lake Tanganyika specifically. Please clarify, and, if the latter, how does this chime with line 179pp where your BNF rates are described as not particularly high, i.e. somewhere in the mid to lower range compared to other lakes?

We agree with the reviewer that this statement was not clear. Our intention was to provide a possible explanation as to why the highest N fixation rates within oligotrophic water bodies - including our case in Lake Tanganyika - (but not all water bodies in general) are often associated with bloom-forming cyanobacteria. However, we have since dropped this line in the revised manuscript, in order to streamline the discussion.

Methods

I could not find information on the BNF rate detection limits anywhere. Especially, since they are invoked in the results, please provide them.

We agree with the reviewer that the background information about our established limits of detection (LOD) was missing. In the revised manuscript, we have provided additional information regarding how N₂ fixation rates were calculated (e.g., LOD, error, AN₂, PN atom%), which is now described in the methods on lines 439-446, and shown in detail in the newly added supplementary Tables S7-S9 (also see response to reviewer 2 for more details). In the methods section we describe the limits of detection as follows, “Calculated limits of detection ($LOD = \delta^{15}N_{control} + 3 * standard\ deviation[\delta^{15}N_{standards}]$), were unusually low with $3 * standard\ deviation[\delta^{15}N_{standards}] < 0.4\ ‰$ in all batches (Tables S7-S9). Thus, we conservatively defined the LOD as $\delta^{15}N_{control} + 4\ ‰$ (minimum change; Montoya et al. 1996). We furthermore defined samples as <LOD, if their $\delta^{15}N$ values or calculated N₂ fixation rates were negative (Tables S7-S9). Standard error was calculated from all samples of one depth. The $\delta^{15}N$ values of the standards showed no significant linear trend within all analyzed batches (Spearman rank correlation test, $p > 0.05$). PN mass on filters ranged from 12-79 $\mu g\ N$ for treatments and 11-62 $\mu g\ N$ for controls (Tables S7-S9).”

Line 506: there should be no comma after “Callbeck et al.”

Done.

Supplementary Discussion

In the “Effect of N₂ fixation on nitrogen loss” section you write that the correlation between N-removal and N-fixation rates suggest that N-loss may be driven by N-fixation. However, given that N-fixation rates \ll N-removal rates (correct?), wouldn't you expect the causal error in the opposite direction? How can a tiny amount of N-fixation drive large rates of N-loss? Since more N-fixation occurs in the stably stratified regions, isn't it also possible (more likely?) that the stable stratification and resulting anoxia allow for more (anaerobic) N-removal?

We thank the reviewer for raising this interesting discussion. To clarify, we see a positive correlation between N₂ fixation and N loss across Lake Tanganyika (Fig. S8). In the main manuscript on lines 256-260, we suggested that the sinking of newly fixed organic N helps to fuel N loss in the anoxic zone, and drive especially high rates in the northern ACM-dominated basin relative to the ACM-void southern basin. Overall, the export of fixed N from N₂ fixation amounts to roughly 12% and 7% of N loss in the North and South, respectively (Table 1). Alternatively, as the reviewer suggested, we could also read Fig. S8 in the opposite direction given that N loss outpaces N₂ fixation (i.e., “causal error in the opposite direction”). However, given that the system is largely stratified (with the exception of the upwelling in the South), how specifically N loss in the anoxic water column influences N₂ fixation in surface waters is unclear. Along

those lines, we see little evidence of either denitrification or anammox greatly imprinting the $\delta^{15}\text{N}$ of DIN, and in turn PON, in upper waters, even in those parts of the lake where upwelling occurred (see also response below for more details). Upwelling of nutrients into surface waters only carried regenerated nitrate from approximately 100 m depth, but not of deeper, denitrified waters (Fig. 1c). Hence, we currently subscribe to the idea that the sinking of organic N (derived from N_2 fixation), while small, contributed to the higher rates of N-loss, especially in the North compared to the South basin (Fig. S8).

Reviewer #2 (Remarks to the Author):

SUMMARY

This article presents a comprehensive study of N_2 fixation and diazotroph biogeography in a large model lake. The authors analyzed standard hydrographic parameters including nutrients, N_2 fixation rates, denitrification rates, and metagenomes. The major findings are as follows:

1. Upwelling affects the distribution of surface N_2 fixation and diazotrophic filamentous cyanobacteria.
2. Surface N_2 fixation in the northern and central basin, where resupply of dissolved inorganic N via upwelling is low, likely fuels new production.
3. N_2 fixation rates within deep anoxic waters were “low but persistent” [line 246].
4. Diazotrophs displayed clear biogeographic patterns, with chemotactic heterotrophic (putative) diazotrophs dominating in upwelling waters.

RESEARCH CONTEXT AND REVIEWER PERSPECTIVE

Understanding the role of physico-chemical dynamics in regulating aquatic N_2 is essential to closing N budgets, on both local and global scales, and predicting future change. Controls on N_2 fixation remain poorly constrained, largely due to the metabolic diversity of extant diazotrophs, and represent a major area of uncertainty in our understanding of the global N cycle and, due to the importance of N limitation to aquatic productivity, the carbon cycle as well.

This study offers insight into N cycling in a model lake—an aquatic system in which N dynamics are currently under-studied. While the role of the major hydrodynamic phenomenon discussed here, upwelling, in regulating N_2 fixation has been the center of a lot of attention in marine systems, the diazotrophs which this article reports from Lake Tanganyika are uncommon in most marine upwelling systems. Consequently, I believe this article provides an interesting perspective and will be a useful addition to the body of literature on the topic.

On the whole, I found the article to be articulate and the study to be comprehensive. Below, I list some specific comments which I believe will enhance the article if addressed. Excepting only the first of my major comments, I do not expect any changes which I suggest to alter the major findings of the article.

We greatly appreciate the reviewer’s support of the work.

MAJOR COMMENTS

- The authors do not discuss how N_2 fixation rate detection limits were assessed (or how rates and rate error was calculated). This is a major omission that has the potential to change major point #3 listed above. How to assess the detectability of N_2 fixation has been a major issue within the community in the past five years (see White et al. 2020, cited in the MS). “Low but persistent” rates have been increasingly reported from deep waters; however, in many cases these may represent methodological artifacts (see Selden et al. 2021 doi: 10.1002/lno.11735). Low rates, when averaged across the large volume of the world’s deep waters, could significantly alter the global N budget (see Benavides et al 2018 doi: 10.3389/fmars.2018.00108 for discussion of the potential significance), so please consider them carefully.

We agree with the reviewer that the background information regarding the limit of detection (LOD) was missing. In the revised manuscript, we have provided additional information on how N₂ fixation rates were calculated (e.g., LODs, error, AN₂, PN atom%), which is now described in the methods section on lines 439-446, and shown in detail in the newly added supplementary Tables S7-S9. In the methods section we describe the limits of detection as follows, “Calculated limits of detection ($LOD = \delta^{15}N_{control} + 3 * standard\ deviation[\delta^{15}N_{standards}]$), were unusually low with $3 * standard\ deviation[\delta^{15}N_{standards}] < 0.4 ‰$ in all batches (Tables S7, S8, and S9). Thus, we conservatively defined the LOD as $\delta^{15}N_{control} + 4 ‰$ (minimum change; (Montoya et al. 1996)). We furthermore defined samples as <LOD, if their $\delta^{15}N$ values or calculated N₂ fixation rates were negative (Tables S7, S8 and S9). Standard error was calculated from all samples of one depth. The $\delta^{15}N$ values of the standards showed no significant linear trend within all analyzed batches (Spearman rank correlation test, $p > 0.05$). PN mass on filters ranged from 12-79 μg N for treatments and 11-62 μg N for controls.”

Based on the definition of LOD above, the rates of N₂ fixation measured in the anoxic zone were above the limit of detection and therefore not considered methodological artifacts. We thank the reviewer for raising this discussion about LOD.

• Similarly, to facilitate downstream use of the N₂ fixation rate data, the authors should report all values used to calculate rates (and all rates in tabular format)—either in a data repository or in a supplemental table. Best data practices are discussed in White et al 2020.

We agree with the reviewer that providing relevant information for reproducibility is necessary. We have opted to include this information in a supplementary table for our three Lake Tanganyika campaigns (2017, 2018 and 2019), which can be found in the newly added Tables S7, S8, and S9, respectively. We have included a sample table below, the full table with all sampled stations can be found in Table S8.

SAMPLE Table S8: Methodological parameters for determining N₂ fixation rates during the Apr/May 2018 campaign. SD standards: standard deviation of all internal standards during the respective mass spectrometer run. The calculated limit of detection (LOD) was defined as $\delta^{15}N = \delta^{15}N_{control} + 3 * standard\ deviation(\delta^{15}N_{standards})$. The applied LOD was conservatively defined as $\delta^{15}N_{control} + 4 ‰$ (minimum change; (Montoya et al. 1996)). Treatment samples >LOD are indicated in dark green and treatment samples <LOD in light green.

station	sample	depth	batch	$\delta^{15}N$ ‰	atom % sample	mass N μg	atom % ¹⁵⁻¹⁵ N ₂ (aq.) %	LOD (calculated) ‰	$\Delta\delta^{15}N$ (sample-LOD) ‰	$\Delta\delta^{15}N$ (sample-control) ‰	volume filtered L	incubation time d	N fixation rate nM N d ⁻¹	
	1	1	190111	16.2	0.3779	57.2	3.57	-0.4	16.6	17.7	4.57	1.00	1.5	
	2		190321	23.5	0.3800	46.1	4.37	-0.7	24.3	25.0	4.57	1.00	1.4	
	1	20	190111	77.1	0.4007	51.9	4.95	-2.4	79.5	80.5	4.57	1.00	5.5	
	2		190321	168.1	0.4359	54.3	5.16	-2.7	170.8	171.5	3.83	1.02	13.7	
	1	50	190111	11.3	0.3749	34.6	4.71	1.9	9.4	10.5	3.74	1.03	0.6	
	2		190321	8.5	0.3738	50.8	6.15	1.6	7.0	7.7	4.57	1.04	0.4	
	1	144	190111	24.5	0.3799	26.7	3.94	8.4	16.1	17.1	4.57	1.03	0.8	
	2		190321	24.5	0.3792	28.5	4.23	8.1	16.4	17.1	4.57	1.05	0.7	
	2	1	156	190111	-6.8	0.3666	30.5	3.62	-8.7	1.9	2.9			
		2		190321	-1.1	0.3699	42.6	4.33	0.7	-1.8	8.6			
			156	190321	-9.8	0.3669	37.1							
			144	190321	7.4	0.3728	27.5							
		control	20	190321	-3.4	0.3696	49.9							
			1	190321	-1.4	0.3725	56.4							
		50	190321	0.9	0.3708	56.4								
	SD standards		190111	0.3	0.0001									
			190321	0.2	0.0005									

• I recommend that the authors revisit the references in the introduction. There are a few points where the inclusion of key references may change the authors’ statements. See my comments below. We appreciate the reviewer’s attention to detail and his/her efforts to provide additional context for the statements in the introduction. In most, if not all the comments below, we have clarified or added additional citations according the

reviewer's suggestion in the revised manuscript. We feel this has helped to improve the accuracy of statements in the introduction. In addition, we have made significant efforts to streamline the introduction in the revised manuscript (based on the general advice of Reviewer 1), which is now roughly half the size as compared to the original version.

MINOR COMMENTS

- [line 43] These seemed like an odd assortment of references to choose, and why are they only listed for P? The potential for Fe limitation of N₂ fixation was first proposed by Falkowski. Fixed N as a control is reviewed by Knapp 2012, who cites original literature on the topic (from the late '90s, early '00s).

We thank the reviewer for their attention to detail. Indeed, Falkowski and Knapp represent key open ocean studies in this respect and subsequently were added to the revised manuscript on line 39. The assortment of studies reflects not only open ocean systems, but also extends to lakes and basins.

- [47] DIN species as a control differs by diazotroph. I recommend citing papers from the genetics literature where, for common taxa, regulatory pathways are well-studied (eg Dixon & Kahn 2004, Nat Rev Microbiol).

We thank the reviewer for the suggestion, we have therefore added the reference by Dixon & Kahn 2004 on line 42, which provides a more molecular-based perspective of nitrogen fixation controls.

- [60-61] Regarding upwelling references and the “5 nmol L⁻¹ d⁻¹” value – These papers are peculiar choices for several reasons. First, there may have been methodological issues with the cited studies (discussed in Selden et al 2021 doi: 10.1002/lno.11735). Second, there have been numerous studies from other coastal upwelling systems like the eastern tropical north pacific where rates can be on the order of 10-100 nmol N L⁻¹d⁻¹ (eg Turk-Kubo et al 2021 doi: s43705-021-00039-7; Selden et al 2019 doi: 10.1029/2019GB006242).

We thank the reviewer for highlighting the study by Selden et al., 2021, indeed, this study provides a nice collation of nitrogen fixation rates, along with additional nuances regarding specific measured values. In the revised manuscript, we have now dropped the 5 nmol L⁻¹ d⁻¹ cutoff value to describe low rates and instead have re-written the statement to read as follows (lines 54-59):

“Field studies, however, have found upwelling N₂ fixation rates to be overall lower than in much of the open ocean (e.g. Selden et al. 2021 and references therein). While rates vary widely from below the limit of detection to as high as 100 nM d⁻¹ (Fernandez et al. 2011; Dekaezemacker et al. 2013; Luo et al. 2014; Knapp et al. 2016; Bonnet et al. 2017; Selden et al. 2019, 2021; Wang et al. 2019; Turk-Kubo et al. 2021; Kittu et al. 2023), they do not re-supply the N lost prior to/during upwelling as predicted by previous biogeochemical models (Deutsch et al. 2007). More recent estimates indicate that N₂ fixation contributes little as a ‘new’ N source in upwelling regions, contributing only ~ 1% to the export of organic N and carbon (Wang et al. 2019).”

- [64] Decoupling due to Fe limitation is likely more important (Weber & Deutsch 2014 www.pnas.org/cgi/doi/10.1073/pnas.1317193111).

We agree with the reviewer that iron limitations are a key constraint on N₂ fixation rates and have therefore added the following (on lines 47-48): “However, absolute rates of N₂ fixation in these regions can vary by orders of magnitude (Luo et al. 2012), and are possibly linked to varying fluxes of P and Fe into these ecosystems (Weber and Deutsch 2014).

- [66] What do the authors consider to be “hotspots”? To my mind, hotspots are where rates are exceptionally high. 5 nmol N L⁻¹ d⁻¹ is not exceptionally high compared to coastal waters where even low N₂ turnover comes to be a high rate due to high biomass (eg in Mid-Atlantic Bight -- Mulholland et al. 2019 GBC; Selden et al 2021 doi: 10.1002/lno.11727; Tang et al 2020 [10.1038/s41396-020-0703-6](https://doi.org/10.1038/s41396-020-0703-6)).

We acknowledge that the word “hotspot” was rather vague. In this particular statement we were attempting to emphasize the recent global ocean modelling work by Wang et al. 2019. Wang et al., 2019, highlight that in ocean gyre systems

50% of organic nitrogen export rates are derived from biological nitrogen fixation, which drops to only 1% in upwelling regions (see their Fig. 4). In this respect, ocean gyres are important sites of N₂ fixation especially relative to upwelling systems. However, we understand that this statement may be considered as vague and lacked context, which we have now provide in the revised manuscript on lines 45-47. It now reads as “The stratified, oligotrophic open oceans, such as the subtropical ocean gyres, are typically important sites of N₂ fixation, driving up to 50% of organic N and carbon export rates (Wang et al. 2019).”

- [71] 5 nmol N L⁻¹ d⁻¹ is not that high compared to N₂ fixation rates in high biomass coastal waters. See refs above, among others.

We have since dropped this reference to a specific value of N₂ fixation in favor of a more general description of its importance across the global ocean, please also see comments above.

- [82] The ecophysiology of N₂ fixation by non-cyanobacterial autotrophic diazotrophs is not well understood. Consider rephrasing.

We have dropped this line in the revised manuscript in order to provide a more streamline introduction.

- [145] I recommend adding a sentence here noting that nitrate was almost completely drawn down below ~150 m (due to denitrification or anammox?)

We agree with the reviewer and have subsequently added the following sentence (see lines 108-110), “Nitrate concentrations eventually fell below detection from 150 m downward, due to anammox and denitrification activity (Callbeck et al. 2021b, see also Supplementary Discussion).”

- [184] Ref 29 is inappropriate here—not an upwelling study

We have removed the reference.

- [198-212] Can the authors comment as to whether denitrification/anammox may have affected δ¹⁵N observed in upper waters, particularly where denitrified waters are upwelled?

We think that the degree to which waters are upwelled plays a role in our answer here. The nitrate N isotope data indicate low-δ¹⁵N values for regenerated nitrate within the nitrate concentration maximum, and more elevated nitrate δ¹⁵N values in the water column regions below, which have undergone partial denitrification (Fig. S4 and S7a). Wind-driven upwelling is a feature of the south basin. During our study period, the upwelling is not intense enough to push denitrified waters to the euphotic zone, though. If this were the case, we would expect to observe lower temperatures (~25°C) and oxygen levels in the upper water layers. If partially denitrified waters affected the δ¹⁵N signals in the euphotic zone, we would also expect the PON to be more enriched in ¹⁵N. For example, in a situation where nitrate utilization is quantitative, the δ¹⁵N of the PON directly reflects the δ¹⁵N of the DIN source, which would, in the case of partially denitrified nitrate pool be tagged by elevated δ¹⁵N. If at all, partially denitrified waters played only a subordinate role for N assimilation in the photic zone. More likely, low-δ¹⁵N nitrate from the nitrate maximum and ammonium was assimilated. The deep water ammonium pool carries a particularly “light” isotopic composition (~ -0.4 ‰; Fig. S4).

As for anammox – the main nitrogen removal pathway in Lake Tanganyika (Callbeck et al. 2021b; Fig. S7b) – the answer is a bit more complex. First of all, the anammox nitrite/nitrate isotope signature comprises multiple overlying kinetic and equilibrium isotope effects, which have been verified only in enrichment culture (Brunner et al. 2013). The expression under natural condition, and under substrate limitation is unclear. Moreover, it is difficult to say to what extent nitrite oxidation to nitrate, with an inverse N isotope effect, has contributed to the subsurface nitrate pool, and in turn to its N isotopic signature. As for the ammonium N isotope effect, one would expect fractional ammonium oxidation by anammox, as is the case with aerobic ammonium oxidation, to result in significant ¹⁵N enrichment in the residual ammonium pool, which, however was not evident from the observational data (Fig. S4).

In summary, we would argue that neither denitrification nor anammox have greatly affected the δ¹⁵N of DIN, and in turn PON, in upper waters, even in those parts of the lake where upwelling occurred. Upwelling of nutrients into surface waters only carried regenerated nitrate from approximately 100 m depth, but not of deeper, denitrified waters (Fig. 1c).

- [242] These are not recent ideas

We agree with the reviewer and have dropped the word “recent” in the revised manuscript (line 228).

- [246] Per my comment above, please consider detection and quantification limits very carefully before asserting that rates are “low but persistent” in deep waters. Unless [PN] is very very low, I am dubious that <1 nmol N-1 d-1 can be distinguished from noise with high confidence.

We agree with the reviewer that the background information about our established LODs was missing. This information is now included (see reply above including newly added Tables S7-S9). In brief, we defined the LOD as the commonly used minimum change of 4 ‰ between treatment and control sample (Montoya et al., 1996). We furthermore defined samples as $<LOD$, if their $\delta^{15}N$ values or calculated N_2 fixation rates were negative (Tables S7, S8 and S9). Based on the above definition of LOD, the rates of N_2 fixation measured in the anoxic zone (< 1 nmol $L^{-1} d^{-1}$) were above the limit of detection and therefore not considered methodological artifacts. N fixation rates <1 nM $N d^{-1}$ were often reported from oligotrophic environments (Benavides et al. 2011; Halm et al. 2012; Dekaezemacker et al. 2013). In the revised manuscript we have since dropped “low, but persistent”; the sentence now reads as follows (line 232), “We also observed rates of N_2 fixation in the oxygen-deficient waters of Lake Tanganyika.”

- [272] Can surface N_2 fixation make up for fixed N loss below?

The export of newly-fixed N from filamentous cyanobacteria in surface waters could contribute to 12% of the fixed N lost in the northern oxygen-deficient zone (Table 1). This information is now indicated on lines 254-256 in the revised manuscript.

- [282] *nifH* alone does not encode a functioning nitrogenase. Please make note of this somewhere so that readers can understand why the difference between *nifH*-encoding metagenomes and *nifHDK*-encoding metagenomes is important.

We agree with the reviewer that this point needed to be better emphasized. We have made this clear on lines 281-283, where it now reads as, “Pseudo-*nifH* sequences, which can comprise a relatively large fraction of curated *nifH* databases, may not be necessarily representative of *bona fide* diazotrophs, as they are unlikely to encode for a functional nitrogenase (Kazumori et al. 2022).” We have made additional changes to the organization of the ideas according to the Reviewer 1 (see lines 286-288). We believe the combination of these changes help to emphasize the point raised by the reviewer above.

- Did *nifHDK*-encoding MAGs also have the genes for the biosynthetic pathway ie *nifENB*?

We did identify *nifENB* gene sequences in the metagenomes of *Dolichospermum* (Table S3), *Chlorobium* (Table S4), *Aquabacterium* (Table S5), and *Pseudomonas* (Table S6) – the key species in this study. This information was included in the original manuscript in the supplementary tables, but we have now clarified that a complete list of accessory nitrogen fixation genes is detailed elsewhere (Table S3-S6) in the main text of the revised manuscript (lines 283-286). These tables include not only *nifENB*, but also other *nif*-related genes (e.g., *nifQOXYZM*).

- [296-303] Fe-only and V-encoding nitrogenases almost always co-occur with Mo-nitrogenase (Poudel et al. 2018 <https://doi.org/10.1128/JB.00757-17>). Is it possible that both were present?

At present, we have not done an extensive survey of Fe-only and V-encoding nitrogenases in the metagenomes recovered from Lake Tanganyika. The current manuscript focuses on the molybdenum-iron (MoFe) nitrogenase genes (*nif*), and, while it would be interesting to know if other nitrogenase genes are encoded, we believe this would expand beyond the scope of the current manuscript. The manuscript is already fairly dense with respect to methods and analyses, and instead we have decided to shorten this paragraph (also according to the advice by Reviewer 1) to save room for other key aspects in the discussion. Nevertheless, we appreciate the discussion raised by the reviewer.

- [352] Is the C:N of the diazotrophs found here known?

The C:N ratio of the diazotrophs is not known, we have therefore decided to remove this sentence in the revised manuscript.

- [469] How were anoxic N₂ fixation incubations sampled i.e., did the authors use standard techniques for sampling low-O₂ waters (overflowing 3x, no bubbles, etc)?

This information is now specified on lines 407-409: “We used a gas-tight hose to fill the bottles bubble-free. Anoxic samples were filled with an overflow of approximately the bottle volume to minimize contamination with ambient air.” Overflowing 3x was not possible due to the total available sample volume. We also acknowledge in the original discussion (lines 457-460) that, “...we cannot completely exclude that trace oxygen contamination may have inhibited N₂ fixation by strict anaerobes, such as *Chlorobium*, which dominate in the ACM (discussed below). Therefore, the rates measured in the anoxic zone likely represent the lower limit of N₂ fixation activity in Lake Tanganyika.”

- [477] Are the “membranes” used here different from the standard serum caps used with PETG incubation bottles?

No, we used the Thermo Scientific™ Nalgene™ Autoclavable Septum Closure, which we believe is the standard.

- [481] How did the authors establish that stocks were ammonium-free?

We agree with the reviewer that we cannot establish the absence of ammonium with certainty. The remark was related to the cited study of Dabundo et al. 2014, which showed that ¹⁵⁻¹⁵N₂ from some companies was systematically contaminated with ammonium and/or nitrate. We thus removed “ammonium-free” from the manuscript (line 413).

- [483] Why not use a syringe to remove the bubble? Did the authors open even anoxic incubations? Can the authors comment on whether the introduction of new air may have affected the dissolved [N₂] concentration i.e., could dissolved ¹⁵N have decreased as it equilibrated with this new light N air?

Concerning the first comment: We agree with the reviewer that using a syringe would have reduced the potential for air exchange. We applied the same methodology for all samples (except for using glass bottles for anoxic samples). As long as bottles are only opened briefly without any turbulence, N₂ does not equilibrate rapidly, which was the fundamental issue of the ¹⁵N₂ bubble method, as N₂ is inherently much less soluble than oxygen (now added on lines 418-420). Nonetheless, on lines 447-456, we now elaborate why we think that our rates should be regarded as conservative estimates. We qualify our arguments with the likely effects of introducing atmospheric oxygen and nitrogen. In the revised manuscript, this part now reads as follows:

“Trace oxygen contamination in anoxic incubations was unavoidable, and may have biased N₂ fixation rates in those samples. Assuming that an enhanced activity of aerobic diazotrophs in anoxic samples through the introduced amounts of oxygen was the dominant effect, we would expect to measure the highest N₂ fixation rates in samples from the anoxic zone in the south, where facultative aerobic diazotrophs, such as *Aquabacterium* or *Pseudomonas*, were more abundant than in the north (Figs. 2 and 3). By contrast, determined N₂ fixation rates in anoxic samples were lower in the south than in the north (Figs. 1f, 2a, and S5). Moreover, during the collection of the labelling atom percent subsample the introduction of atmospheric N₂ may have lowered the actual ¹⁵⁻¹⁵N₂ concentration compared to the measured value. Hence, N₂ fixation rate estimates should be considered conservative”

- [483] What do the authors mean by “shaking”? Most people gently roll incubation vials to avoid rupturing cells.

We thank the reviewer for highlighting this, indeed, the bottles were treated gently. The word “shake” was misleading. In the revised manuscript, it now reads as (lines 414-415), “The labelled samples were equilibrated for approximately 30 minutes by rolling/inverting the bottles.”

- [489] What was this average value? Please report all data used to calculate rates for all incubations in a table somewhere, following best data practices in N₂ fixation reporting (see White et al 2020).

Done, see comment below and newly added Table S7-S9.

- [503] Was PN at t=0 collected? How were detection limits calculated? How was error

calculated? Was AN2 measured for each incubation (apart from in 2019)? What was the range of those values? What was the PN mass on the filters used to get PN atom% values? What were the linearity criteria used in this study? How were those criteria established? There is a lot of missing information here. It could all go in supplemental text, but needs to be included. Please revisit White et al 2020.

We agree with the reviewer that providing relevant information for reproducibility is necessary. The indicated parameters are now listed in Tables S7-S9 and integrated in the methods section in the manuscript:

LOD: Calculated limits of detection ($LOD = \delta^{15}N_{\text{control}} + 3 * \text{standard deviation}[\delta^{15}N_{\text{standards}}]$), were unusually low with $3 * \text{standard deviation}[\delta^{15}N_{\text{standards}}] < 0.4 \%$ in all batches (Tables S7-S9). Thus, we conservatively defined the LOD as $\delta^{15}N_{\text{control}} + 4 \%$ (minimum change; Montoya et al. 1996). We furthermore defined samples as <LOD, if their $\delta^{15}N$ values or calculated N_2 fixation rates were negative (Tables S7-S9).

ERROR: Standard error was calculated from all samples of one depth.

AN2: Atom percent of dissolved $^{15-15}N_2$ in treatment samples was measured for each incubation bottle individually. In 2018, values ranged from 4-10 % in most cases, with extreme outliers between 0.3 and 11.3 % (Tables S7-S9).

PN atom%: PN mass on filter ranged from 15-79 $\mu\text{g N}$ for treatments and 11-56 $\mu\text{g N}$ for controls (Tables S7-S9).

Linearity criteria: The $\delta^{15}N$ values of the standards showed no significant linear trend within all analyzed batches (Spearman rank correlation test, $p > 0.05$).

PN at t0: Calculating this value was not possible. The natural abundance isotope samples were analyzed in different batches, and hence, the standards were not arranged in a wide mass range to derive the PN mass of the samples.

• [551] How long did the filters sit before freezing? Was this within limits recommended by the manufacturer?

The filters sat between 1 and 2 weeks before freezing. We found no limits provided by the manufacturer. It reads as follows in the revised manuscript (lines 493-494), “Briefly, lake water was filtered onto 0.2 μm cellulose acetate filters, fixed with RNAlater (Sigma Life Science) and stored at 4 °C for 1-2 weeks”

References cited in the response to reviewer comments:

- Benavides, M., N. Agawin, J. Aristegui, P. Ferriol, and L. Stal. 2011. Nitrogen fixation by *Trichodesmium* and small diazotrophs in the subtropical northeast Atlantic. *Aquat. Microb. Ecol.* **65**: 43–53. doi:10.3354/ame01534
- Bergamino, N., S. Horion, S. Stenuite, Y. Cornet, S. Loisel, P.-D. Plisnier, and J.-P. Descy. 2010. Spatio-temporal dynamics of phytoplankton and primary production in Lake Tanganyika using a MODIS based bio-optical time series. *Remote Sens. Environ.* **114**: 772–780. doi:https://doi.org/10.1016/j.rse.2009.11.013
- Bonnet, S., M. Caffin, H. Berthelot, and T. Moutin. 2017. Hot spot of N_2 fixation in the western tropical South Pacific pleads for a spatial decoupling between N_2 fixation and denitrification. *Proc. Natl. Acad. Sci. U. S. A.* **114**: E2800–E2801. doi:10.1073/pnas.1619514114
- Callbeck, C. M., D. E. Canfield, M. M. M. Kuypers, P. Yilmaz, G. Lavik, B. Thamdrup, C. J. Schubert, and L. A. Bristow. 2021a. Sulfur cycling in oceanic oxygen minimum zones. *Limnol. Oceanogr.* **66**: 2360–2392. doi:https://doi.org/10.1002/lno.11759
- Callbeck, C. M., B. Ehrenfels, K. B. L. Baumann, B. Wehrli, and C. J. Schubert. 2021b. Anoxic chlorophyll maximum enhances local organic matter remineralization and nitrogen loss in Lake Tanganyika. *Nat. Commun.* **12**. doi:10.1038/s41467-021-21115-5

- Dabundo, R., M. F. Lehmann, L. Treibergs, C. R. Tobias, M. A. Altabet, P. H. Moisander, and J. Granger. 2014. The Contamination of Commercial $^{15}\text{N}_2$ Gas Stocks with ^{15}N -Labeled Nitrate and Ammonium and Consequences for Nitrogen Fixation Measurements J.B. Love [ed.]. *PLoS One* **9**: e110335. doi:10.1371/journal.pone.0110335
- Dekaezemacker, J., S. Bonnet, O. Grosso, T. Moutin, M. Bressac, and D. G. Capone. 2013. Evidence of active dinitrogen fixation in surface waters of the eastern tropical South Pacific during El Niño and La Niña events and evaluation of its potential nutrient controls. *Global Biogeochem. Cycles* **27**: 768–779. doi:10.1002/gbc.20063
- Deutsch, C., J. L. Sarmiento, D. M. Sigman, N. Gruber, and J. P. Dunne. 2007. Spatial coupling of nitrogen inputs and losses in the ocean. *Nature* **445**: 163–167. doi:10.1038/nature05392
- Ehrenfels, B., M. Bartosiewicz, A. S. Mbonde, and others. 2021. Diazotrophic cyanobacteria are associated with a low nitrate resupply to surface waters in Lake Tanganyika. *Front. Environ. Sci.* **9**: 277. doi:10.3389/fenvs.2021.716765
- Falkowski, P. G. 1997. Evolution of the nitrogen cycle and its influence on the biological sequestration of CO_2 in the ocean. *Nature* **387**: 272–275. doi:10.1038/387272a0
- Farnelid, H., M. Bentzon-Tilia, A. F. Andersson, S. Bertilsson, G. Jost, M. Labrenz, K. Jürgens, and L. Riemann. 2013. Active nitrogen-fixing heterotrophic bacteria at and below the chemocline of the central Baltic Sea. *ISME J.* **7**: 1413–1423. doi:10.1038/ismej.2013.26
- Fernandez, C., L. Fariás, and O. Ulloa. 2011. Nitrogen Fixation in Denitrified Marine Waters J.A. Gilbert [ed.]. *PLoS One* **6**: e20539. doi:10.1371/journal.pone.0020539
- Hallström, S., J. B. Raina, M. Ostrowski, and others. 2022. Chemotaxis may assist marine heterotrophic bacterial diazotrophs to find microzones suitable for N_2 fixation in the pelagic ocean. *ISME J.* **16**: 2525–2534. doi:10.1038/s41396-022-01299-4
- Halm, H., P. Lam, T. G. Ferdelman, and others. 2012. Heterotrophic organisms dominate nitrogen fixation in the South Pacific Gyre. *ISME J.* **6**: 1238–1249. doi:10.1038/ismej.2011.182
- Halm, H., N. Musat, P. Lam, and others. 2009. Co-occurrence of denitrification and nitrogen fixation in a meromictic lake, Lake Cadagno (Switzerland). *Env. Microbiol.* **11**: 1945–1958. doi:10.1111/j.1462-2920.2009.01917.x
- Jayakumar, A., and B. B. Ward. 2020. Diversity and distribution of nitrogen fixation genes in the oxygen minimum zones of the world oceans. *Biogeosciences* **17**: 5953–5966. doi:10.5194/bg-17-5953-2020
- Kazumori, M., M. Yoko, S. Keishi, I. Hideomi, and T. S. Green. 2022. Undervalued Pseudo-nifH Sequences in Public Databases Distort Metagenomic Insights into Biological Nitrogen Fixers. *mSphere* **6**: e00785-21. doi:10.1128/msphere.00785-21
- Kittu, L. R., A. J. Paul, M. Fernández-méndez, and M. J. Hopwood. 2023. Coastal N_2 Fixation Rates Coincide Spatially With Nitrogen Loss in the Humboldt Upwelling System off Peru Global Biogeochemical Cycles. doi:10.1029/2022GB007578
- Knapp, A. N. 2012. The sensitivity of marine N_2 fixation to dissolved inorganic nitrogen. *Front. Microbiol.* **3**: 374. doi:10.3389/fmicb.2012.00374
- Knapp, A. N., K. L. Casciotti, W. M. Berelson, M. G. Prokopenko, and D. G. Capone. 2016. Low rates of nitrogen fixation in Eastern Tropical South Pacific surface waters. *Proc. Natl. Acad. Sci. U. S. A.* **113**: 4398–4403. doi:10.1073/pnas.1515641113
- Langenberg, V. T., S. Nyamushahu, R. Roijackers, and A. A. Koelmans. 2003. External nutrient sources for Lake Tanganyika. *J. Great Lakes Res.* **29**: 169–180. doi:10.1016/S0380-1330(03)70546-2
- Luo, Y. W., S. C. Doney, L. A. Anderson, and others. 2012. Database of diazotrophs in global ocean: Abundance, biomass and nitrogen fixation rates. *Earth Syst. Sci. Data* **4**: 47–73. doi:10.5194/essd-4-47-2012
- Luo, Y. W., I. D. Lima, D. M. Karl, C. A. Deutsch, and S. C. Doney. 2014. Data-based assessment of environmental controls on global marine nitrogen fixation. *Biogeosciences* **11**: 691–708.

doi:10.5194/bg-11-691-2014

- Ma, J., K. L. French, X. Cui, D. A. Bryant, and R. E. Summons. 2021. Carotenoid biomarkers in Namibian shelf sediments: Anoxygenic photosynthesis during sulfide eruptions in the Benguela Upwelling System. *Proc. Natl. Acad. Sci. U. S. A.* **118**. doi:10.1073/pnas.2106040118
- Mills, M. M., C. Ridame, M. Davey, J. La Roche, and R. J. Geider. 2004. Iron and phosphorus co-limit nitrogen fixation in the eastern tropical North Atlantic. *Nature* **429**: 292–294. doi:10.1038/nature02550
- Montoya, J. P., M. Voss, P. Kahler, and D. G. Capone. 1996. A simple high-precision, high-sensitivity tracer assay for N₂ fixation. *Appl. Environ. Microbiol.* **62**: 986–993.
- Naithani, J., and E. Deleersnijder. 2004. Are there internal Kelvin waves in Lake Tanganyika? *Geophys. Res. Lett.* **31**. doi:10.1029/2003GL019156
- O'Reilly, C. M., S. R. Alin, P.-D. Plisnier, A. S. Cohen, and B. A. McKee. 2003. Climate change decreases aquatic ecosystem productivity of Lake Tanganyika, Africa. *Nature* **424**: 766–768. doi:10.1038/nature01833
- Pierella Karlusich, J. J., E. Pelletier, F. Lombard, and others. 2021. Global distribution patterns of marine nitrogen-fixers by imaging and molecular methods. *Nat. Commun.* **12**: 1–18. doi:10.1038/s41467-021-24299-y
- Romero, I. C., N. J. Klein, S. A. Sañudo-Wilhelmy, and D. G. Capone. 2013. Potential trace metal co-limitation controls on N₂ fixation and NO₃⁻ uptake in lakes with varying trophic status. *Front. Microbiol.* **4**. doi:10.3389/fmicb.2013.00054
- Schoffelen, N. J., W. Mohr, T. G. Ferdelman, H. Ploug, and M. M. M. Kuypers. 2019. Phosphate availability affects fixed nitrogen transfer from diazotrophs to their epibionts. *ISME J.* **13**: 2701–2713. doi:10.1038/s41396-019-0453-5
- Selden, C. R., M. R. Mulholland, P. W. Bernhardt, B. Widner, A. Macías-Tapia, Q. Ji, and A. Jayakumar. 2019. Dinitrogen Fixation Across Physico-Chemical Gradients of the Eastern Tropical North Pacific Oxygen Deficient Zone. *Global Biogeochem. Cycles* **33**: 1187–1202. doi:10.1029/2019GB006242
- Selden, C. R., M. R. Mulholland, B. Widner, P. Bernhardt, and A. Jayakumar. 2021. Toward resolving disparate accounts of the extent and magnitude of nitrogen fixation in the Eastern Tropical South Pacific oxygen deficient zone. *Limnol. Oceanogr.* **66**: 1950–1960. doi:10.1002/lno.11735
- Turk-Kubo, K. A., M. M. Mills, K. R. Arrigo, G. van Dijken, B. A. Henke, B. Stewart, S. T. Wilson, and J. P. Zehr. 2021. UCYN-A/haptophyte symbioses dominate N₂ fixation in the Southern California Current System. *ISME Commun.* **1**: 1–13. doi:10.1038/s43705-021-00039-7
- Verburg, P., and R. E. Hecky. 2003. Wind Patterns, Evaporation, and Related Physical Variables in Lake Tanganyika, East Africa. *J. Great Lakes Res.* **29**: 48–61. doi:https://doi.org/10.1016/S0380-1330(03)70538-3
- Villareal, T. A., and E. J. Carpenter. 2003. Buoyancy regulation and the potential for vertical migration in the oceanic cyanobacterium *Trichodesmium*. *Microb. Ecol.* **45**: 1–10. doi:10.1007/s00248-002-1012-5
- Wang, W.-L. L., J. K. Moore, A. C. Martiny, and F. W. Primeau. 2019. Convergent estimates of marine nitrogen fixation. *Nature* **566**: 205–211. doi:10.1038/s41586-019-0911-2
- Wasmund, N., G. Nausch, and M. Voss. 2012. Upwelling events may cause cyanobacteria blooms in the Baltic Sea. *J. Mar. Syst.* **90**: 67–76. doi:10.1016/j.jmarsys.2011.09.001
- Weber, T., and C. Deutsch. 2014. Local versus basin-scale limitation of marine nitrogen fixation. *Proc. Natl. Acad. Sci. U. S. A.* **111**: 8741–8746. doi:10.1073/pnas.1317193111

Reviewer #1 (Remarks to the Author):

I reviewed the original submission of this article and find the revised version clearly improved, and I would like to thank the authors for carefully addressing my concerns. In particular, I find the discussion more concise, well-structured and to the point.

One of my main concerns was addressed by adding the correlation analyses in Figure S9. However, the lack of significant negative correlation between N₂-fix and NO₃-flux as well as 'z_eu-TC' (Figure S9) allows for some lingering doubts regarding the key conclusion. I agree with the authors that the strong correlation between cyanobacteria with both N₂-fixation rates and NO₃-flux helps to address these doubts. And certainly the small number of depth-integrated N₂-fixation datapoints might be to blame for the statistical non-significance. But I think this can be compensated for by a) more detailed/concise correlation analyses, b) additional discussion of Station 2.

a) Unless that's already what they did (in which case I would ask the authors to state that clearly in the caption of Figure S9), I would suggest to use 'euphotic-zone integrated' (rather than 'entire-water-column integrated') N₂-fixation rates in the correlation analysis against NO₃-flux etc. After all, this reflects the actual underlying hypothesis: low NO₃-flux drives N₂-fixation in the euphotic zone (any N₂-fixation contributed from below the euphotic zone is really a confounding influence on this hypothesis test). Alternatively or additionally, nifH abundance and/or d¹⁵N-PON (both within euphotic zone only) could be used as proxies for N₂-fixation in order to address the scarcity of depth-integrated N₂-fixation rates and could be correlated against NO₃ fluxes well. This might further strengthen the case that N₂-fixation is affected by NO₃-flux.

b) If there was one response to my comments on the original version I was unconvinced by, it was the authors' point about the high NO₃-fluxes in Stations 8 and 9 distorting the scale. But the scale is not the problem I was referring to. Rather, the NO₃-fluxes in the southern stations 5-7 are not only not higher, but all of them are lower than those in northern stations 1 and 2. What causes this high NO₃ flux in stations 1 and 2 given that 'z_eu-TC' is very low there and why is there high N₂-fixation in Station 2 despite higher NO₃ flux than in Stations 5-7? My sense is that the authors' conclusion is probably correct for the bulk of the lake but that something else is affecting the NO₃ flux/N₂-fixation relationship in the northernmost stations that needs to be worked out. In effect, Stations 1 and 2 look to me like outliers within the context of the northern basin, and Station 1 is already discussed as such in the newly added lines 215-222: riverine NO₃-input likely causes low cyanobacteria and low N₂-fixation. A similar discussion might be helpful in understanding outlier Station 2: is the elevated NO₃-flux also due to river input, but if so, why is N₂-fixation high, opposing the trend in the rest of the lake?

In summary, I think that despite the low number of depth-integrated N₂-fixation rates, the evidence for the NO₃-flux driving N₂-fixation could be worked out even more clearly by applying some of the suggestions above.

Please find a few more comments below.

Line 175: wouldn't the isotope effect associated with nitrate assimilation be expected to result in a d¹⁵N-PON several per mil lower than the d¹⁵N-NO₃?

Line 178-182: Picking the northernmost and southernmost stations for d¹⁵N-PON comparison in support of the mixing-regime driven differences in N₂-fixation across the lake seems subjective and unnecessarily so. Looking at Figure S4, the trend is indeed very clear across the 5 stations for which d¹⁵N-PON data exist. I would suggest to mention that there are more stations that follow this trend in addition to the two extreme stations. Optionally, a correlation analysis of euphotic zone d¹⁵N-PON vs N₂-fix rates could help to convey this point.

Also, calling Station 3 "the northernmost station" here is a bit confusing. I think it is the northernmost station at which d¹⁵N-PON data is available. Please rephrase accordingly.

Line 238: add superscript "15" to "albeit N₂ incubations were not performed..."

Line 424: using the average labeling percentage from 2018 ("mean_2018_labeling%") for 2019 samples due to supposedly inaccurate labeling percentage data in 2019 seems like a source of error with potentially large consequence on the final rate estimate. Please provide a "sensitivity analysis" to inform the reader how impactful this error might be. For example, the authors could calculate each 2019 N₂-fixation rate with

a) [mean_2018_labeling%] + [1 SD]

b) [mean_2018_labeling%] - [1 SD]

where SD is the standard deviation of the 2018 labeling percentages.

Alternatively, as it seems like the 2019 N₂-fix data are only used in Figure S6 and barely discussed in the main text, consider adding an appropriate warning/caveat ('based on uncertain labeling percentage, see Methods') wherever the 2019 N₂-fix data are used. As a further alternative, it is not clear to me that the 2019 N₂-fixation rate data are really needed at all in this article and the authors might prefer not publishing them given the methodological flaw.

Line 433: I suggest rephrasing to "given that differences were small,..."

Line 456: I would ask the authors to briefly pick up this caveat (rates from anoxic zone being conservative due to trace amounts of O₂) in the Results and Discussion section, since it is common for readers to ignore M&M sections and this seems to be an important qualifier that should not get lost in the Methods section.

Figures/Table

Table 1: please indicate here and/or elsewhere which stations are counted as north/center and which as south.

References 51 and 52: Please subscript the "2" in "N₂".

Supplementary: In several cases (Schubert et al., Callbeck et al.), there is a comma behind 'et al.' that shouldn't be there.

Reviewer #2 (Remarks to the Author):

Manuscript ID: NCOMMS-22-49341A

August 7, 2023

Reviewer 2

General:

I thank the authors for their careful consideration of my initial comments on the MS. I believe the article is much improved and I have only one major point on which clarification is needed (below). I have also included several minor recommendations (associated with line numbers in the revised MS) to aid the authors in polishing their article.

Major comment:

In their reviewer response, the authors indicated the following: "Calculating [PN at t=0] was not possible. The natural abundance isotope samples were analyzed in different batches, and hence, the standards were not arranged in a wide mass range to derive the PN mass of the samples." I understand that the authors cannot provide PN mass and I do not consider this to be an issue. It is very common (and generally reasonable) to assume constant PN concentration over the course of the incubation. However, in systems where denitrification occurs, it is important to measure the natural abundance of ¹⁵N-PN (i.e., the t=0 isotope ratio in the mixing model) and not to assume a value of 0.366 atom-% (as many oceanographic studies do) because denitrification can significantly alter ¹⁵N-PN natural abundance. Failing to consider this can lead to false positives. Can the authors confirm that they used natural abundance ¹⁵N-PN from the same water mass from which N_{fix} incubation water was collected to calculate their N₂ fixation rates? (And my apologies for not clearly articulating the question last time.)

Minor comments:

[22] Remove commas before and after "yet"

[35-37] I recommend placing this sentence as the first of the following paragraph, rather than having it as its own paragraph.

[38] Please replace "may" with "can"

[49] Cyanobacteria are a type of microbe.

[52-55] Upwelling systems in the ETNP and ETSP are relatively Fe-poor and Nfix is likely limited by Fe there. Other upwelling systems (eg the Benguela, see Sohm et al 2011, or California Current System, see Kendra Turk-Kubo's work) can have higher Nfix rates than the ocean gyres.

[63] I, personally, am not convinced that this is "likely" in all aquatic ecosystems (eg <https://www.nature.com/articles/ismej2014119>), but is probably true for pelagic marine systems.

[160-167] What are the errors associated with these estimates?

[188] It may be worth pointing out that Anabaenopsis is also an established N fixer.

[226-228] Filamentous cyanobacterial endosymbionts of diatoms have been found in nutrient-rich waters including river plumes (Amazon – Carpenter et al 1999; Mekong – Grosse et al 2010) and upwelling systems (<https://www.nature.com/articles/s41598-017-18006-5#Fig3>). Personally, I believe that the controls on (filamentous cyanobacterial) diazotrophs differ depending on their whether they are free-living or symbiotic. I recommend specifying "free-living" here.

[254-256] I find the phrasing here a little confusing. I recommend rephrasing so that it is clear that N₂ fixation is not contributing directly to fixed N loss (which is nonsensical) but, rather, is supplying fixed N for denitrification. Also, aren't the denitrified waters being upwelled, supporting a niche for diazotrophs?

[258] Cyanobacteria are part of local production.

[250-260] Do depth-integrated C fixation rates also correlate positively with N loss and N₂ fixation?

[342] P and Fe limitation was not measured in this study. I recommend rephrasing as "not drawn down" instead of "not limiting".

[359] The N demand of these organisms may also be partially met by organic matter degradation i.e., they may not need to fix as much N if its in excess in their (heterotrophic) diet.

[366] The statement after the semi-colon is a fragment, not an independent clause. This is grammatically incorrect.

[365] The articles cited represent a biased sampling of upwelling systems—they are either focused on the ETNP/ETSP or are modeling papers. Other marine upwelling systems have shown evidence of significant N₂ fixation in upwelling systems (<https://www.nature.com/articles/s43705-021-00039-7> and <https://agupubs.onlinelibrary.wiley.com/doi/pdfdirect/10.1029/2005GL025569> and <https://www.nature.com/articles/s41598-017-18006-5> and <https://agupubs.onlinelibrary.wiley.com/doi/pdf/10.1029/2011GL048315>). Nfix in the ETNP and ETSP are probably iron-limited, at least in part, and do not host UCYN-A—a globally significant diazotroph which appears to be favored during upwelling (eg <https://www.frontiersin.org/articles/10.3389/fmars.2022.877562/full>)

We would again like to thank the two reviewers for their support of this work, their attention to detail, and their constructive criticism. Their due diligence has greatly contributed to improving the manuscript.

Below is a point-by-point response to the reviewers' comments, indicated in the red-type face. The grey highlighted line numbers refer to changes made in the revised manuscript (e.g. lines XX-XX). A complete list of references is included at the end of the response to reviewers' comments.

REVIEWER COMMENTS

Reviewer #1 (Remarks to the Author):

I reviewed the original submission of this article and find the revised version clearly improved, and I would like to thank the authors for carefully addressing my concerns. In particular, I find the discussion more concise, well-structured and to the point.

We greatly appreciate the reviewer's constructive feedback.

One of my main concerns was addressed by adding the correlation analyses in Figure S9. However, the lack of significant negative correlation between N₂-fix and NO₃-flux as well as 'z_{eu}-TC' (Figure S9) allows for some lingering doubts regarding the key conclusion. I agree with the authors that the strong correlation between cyanobacteria with both N₂-fixation rates and NO₃-flux helps to address these doubts. And certainly the small number of depth-integrated N₂-fixation datapoints might be to blame for the statistical non-significance. But I think this can be compensated for by a) more detailed/concise correlation analyses, b) additional discussion of Station 2.

We thank the reviewer for raising the discussion regarding correlative analyses. We would like to clarify that while N₂ fixation is not significant at a 95% confidence interval ($p < 0.05$) when correlated against the nitrate flux and Z_{eu} - TC it is however, significant at an 80% confidence interval ($p < 0.2$; see updated S9g). We believe the relatively noisy trend could be attributed to non-steady state conditions, which could be encountered at some stations, like Station 2 (as the reviewer highlighted here and, in their comment, below). For example, a fluctuating DIN supply is expected for Stations 1 and 2, which are positioned near the river inlets of Rusizi and Malagarasi, respectively (discussed in more detail below). These rivers, especially Malagarasi, experience large DIN fluctuations due to rain events during the transition from the wet to the dry season⁵³. Therefore, N₂ fixation rates may suffer from a lagged response by the diazotroph community toward a fluctuating nitrate supply⁴⁴. This could ultimately lead to a wider margin of error in the N₂ fixation rates versus the nitrate flux analysis (Fig. S9g). Overall, we agree with the reviewer that a deeper discussion of Station 2 was needed in order to address the larger correlative issue, we have since added this information above on lines 203-209 (please also see our comment below).

Apart from our short-term ¹⁵-¹⁵N₂ incubations, an even clearer cross-lake pattern emerges if we look at a long-term integrated signature of N₂ fixation based on the $\delta^{15}\text{N-PON}$. Briefly, a low $\delta^{15}\text{N-PON}$ signature (characteristic of N₂ fixation) is observed in the stratified North and increases to more positive values in the upwelling-driven South (Fig. 11). As the reviewer suggested below, we additionally performed correlation analyses of the $\delta^{15}\text{N-PON}_{\text{zeu}}$ in the euphotic zone against the nitrate flux ($R^2 = 0.79$; $p < 0.05$; $N = 5$; see new Fig. S9d), Z_{eu} - TC ($R^2 = 0.61$; $p < 0.05$; $N = 5$; Fig. S9e), and cyanobacteria ($R^2 = 0.63$; $p < 0.05$; $N = 5$; Fig. S9f). And in all cases, we have a significant relationship at a high confidence interval of 95%. Overall, these correlations help to strengthen the underline conclusion that N₂ fixation exhibits a clear North-South trend that varies according to changes in the nitrate flux driven by the upwelling intensity. We have further described these correlations in the revised manuscript on lines 191-193 and 209-211.

Fig. S9 (panels d-f added new): Correlation matrix with the integrated rates of N₂ fixation (0-43 m; N₂ fixation_{Z_{eu}}), the δ¹⁵N of particulate organic nitrogen in the euphotic zone (0-43 m; δ¹⁵N-PON_{Z_{eu}}), the abundance of filamentous, diazotrophic cyanobacteria, the distance between the euphotic depth (Z_{eu}) and the thermocline (Z_{eu} - TC) as well as the estimated nitrate flux (NO₃⁻ flux). A linear regression line is shown with R² and p-values (Spearman rank correlation test) indicated above each panel. Note, that performing a correlation analysis between δ¹⁵N-PON_{Z_{eu}} versus volumetric N₂ fixation rates is not possible because these measurements were not always sampled at the same station (only three stations overlap). Correlative analyses that are statistically significant at a 95-99% (p < 0.05), and 80% confidence interval (p < 0.2) are indicated.

a) Unless that's already what they did (in which case I would ask the authors to state that clearly in the caption of Figure S9), I would suggest to use 'euphotic-zone integrated' (rather than 'entire-water-column integrated') N₂-fixation rates in the correlation analysis

against NO₃-flux etc. After all, this reflects the actual underlying hypothesis: low NO₃-flux drives N₂-fixation in the euphotic zone (any N₂-fixation contributed from below the euphotic zone is really a confounding influence on this hypothesis test). Alternatively or additionally, nifH abundance and/or d¹⁵N-PON (both within euphotic zone only) could be used as proxies for N₂-fixation in order to address the scarcity of depth-integrated N₂-fixation rates and could be correlated against NO₃ fluxes well. This might further strengthen the case that N₂-fixation is affected by NO₃-flux.

To the reviewer's first question regarding Figure S9, yes, we used euphotic-zone integrated N₂ fixation, which has now been clarified in the panels (Fig. S9g, i, and b) as “N₂ fixation_{zeu}”, and in the caption, it now reads as, “Correlation matrix with the integrated rates of N₂ fixation (0-43 m; N₂ fixation_{zeu})...”. We thank the reviewer for pointing this missing information out.

Following the advice of the reviewer, we also performed additional correlative analyses. Namely, we analyzed δ¹⁵N-PON (a long-term integrated signature of N₂ fixation) versus the nitrate flux ($R^2 = 0.79$; $p < 0.05$; $N = 5$; see new Fig. S9d), $Z_{eu} - TC$ ($R^2 = 0.61$; $p < 0.05$; $N = 5$; Fig. S9e), and cyanobacteria ($R^2 = 0.63$; $p < 0.05$; $N = 5$; Fig. S9f), as discussed above. And in all cases, we have a significant relationship at a high confidence interval of 95%. Overall, these correlations help to strengthen the underline conclusion that N₂ fixation exhibits a clear North-South trend that varies according to changes in the nitrate flux driven by the upwelling intensity.

b) If there was one response to my comments on the original version I was unconvinced by, it was the authors' point about the high NO₃-fluxes in Stations 8 and 9 distorting the scale. But the scale is not the problem I was referring to. Rather, the NO₃-fluxes in the southern stations 5-7 are not only not higher, but all of them are lower than those in northern stations 1 and 2. What causes this high NO₃ flux in stations 1 and 2 given that 'z_{eu}-TC' is very low there and why is there high N₂-fixation in Station 2 despite higher NO₃ flux than in Stations 5-7? My sense is that the authors' conclusion is probably correct for the bulk of the lake but that something else is affecting the NO₃ flux/N₂-fixation relationship in the northernmost stations that needs to be worked out. In effect, Stations 1 and 2 look to me like outliers within the context of the northern basin, and Station 1 is already discussed as such in the newly added lines 215-222: riverine NO₃-input likely causes low cyanobacteria and low N₂-fixation. A similar discussion might be helpful in understanding outlier Station 2: is the elevated NO₃-flux also due to river input, but if so, why is N₂-fixation high, opposing the trend in the rest of the lake?

We thank the reviewer for raising this interesting discussion. We suspect that station 2 was likely under non-steady state conditions at the time of sampling due to fluctuating riverine DIN input. In the earlier version of the manuscript, we mentioned that station 1 – a clear outlier to the North-South trend – is influenced by a river called Rusizi. Rusizi is one of the largest and most consistent inputs of DIN into the lake, which has a fairly steady flow rate year-round with only a 1.6-fold difference between wet and dry seasons (Langenberg et al. 2003). However, in the previous version, we overlooked a potential role for a river called Malagarasi to influence Station 2. For Malagarasi, which enters the lake's east side, the flow rate and DIN input can vary markedly by 7-fold between wet and dry seasons, standing in stark contrast to Rusizi (Langenberg et al. 2003). Because our sampling took place during the transition from the wet to the dry season (April-May), it is quite possible that DIN concentrations were in a non-steady state (i.e., exhibiting a strong degree of variability). This, combined with the fact that N₂ fixation may suffer from a lag effect, could explain why residual activity remains even in spite of a moderate nitrate flux. In the revised manuscript, we have added a brief discussion of the potential impact of riverine influence over both stations 1 and 2, which reads as follows (lines 202-211):

“Moreover, the rates of N₂ fixation showed a trend towards higher activity with a decreasing nitrate flux, albeit this trend was relatively noisy ($R^2 = 0.50$, $p < 0.2$; Fig. S9g). Non-steady state conditions might potentially

contribute to a wider margin of error in correlative analyses at some stations. For example, a fluctuating DIN supply is expected for Stations 1 and 2, which are positioned near the river inlets of Rusizi and Malagarasi, respectively. These rivers experience large DIN fluctuations due to rain events during the transition from the wet to the dry season⁵³. Therefore, N₂ fixation rates may suffer from a lagged response by the diazotroph community toward a fluctuating nitrate supply⁴⁴. Nevertheless, based on the $\delta^{15}\text{N-PON}_{\text{Zeu}}$ signature (a long-term indicator of N₂ fixation) we find a strong correlation with the nitrate flux ($R^2 = 0.79$; $p < 0.05$; Fig. S9d)."

In summary, I think that despite the low number of depth-integrated N₂-fixation rates, the evidence for the NO₃-flux driving N₂-fixation could be worked out even more clearly by applying some of the suggestions above.

We thank the reviewer for the suggestion to do additional correlative analyses. As discussed above, we performed $\delta^{15}\text{N-PON}$ correlations against the nitrate flux ($R^2 = 0.79$; $p < 0.05$; $N = 5$; Fig. S9d), $Z_{\text{eu}} - \text{TC}$ ($R^2 = 0.61$; $p < 0.05$; $N = 5$; Fig. S9e), and cyanobacteria ($R^2 = 0.63$; $p < 0.05$; $N = 5$; Fig. S9f).

Please find a few more comments below.

Line 175: wouldn't the isotope effect associated with nitrate assimilation be expected to result in a $\delta^{15}\text{N-PON}$ several per mil lower than the $\delta^{15}\text{N-NO}_3$?

We believe that a close coupling between organic N remineralization to nitrate (via nitrification; also discussed in Callbeck et al. 2021), and nitrate re-assimilation at the water depth where nitrate accumulates, would not necessarily result in a large N isotope fractionation between the PON and the nitrate pools. Yet the point we want to make here is that the $\delta^{15}\text{N}$ of PON in the surface, where we expect N₂ fixation rates to be highest, is significantly lower, confirming a different N source (i.e., N₂) with respect to the subsurface (i.e., probably nitrate).

In addition, we have dropped the line which followed shortly after which previously stated "The low and often undetectable nitrate concentrations within the surface waters imply complete uptake by phytoplankton. Under these conditions, no fractionation will take place between the substrate (nitrate) and the product (PON) N pools." We believe this was superfluous to the overall message of N₂ fixation.

Line 178-182: Picking the northernmost and southernmost stations for $\delta^{15}\text{N-PON}$ comparison in support of the mixing-regime driven differences in N₂-fixation across the lake seems subjective and unnecessarily so. Looking at Figure S4, the trend is indeed very clear across the 5 stations for which $\delta^{15}\text{N-PON}$ data exist. I would suggest to mention that there are more stations that follow this trend in addition to the two extreme stations. Optionally, a correlation analysis of euphotic zone $\delta^{15}\text{N-PON}$ vs N₂-fix rates could help to convey this point.

We agree with the reviewer that the cross-station trend in $\delta^{15}\text{N-PON}_{\text{Zeu}}$ from North to South is quite clear between Stations 3 and 9 (the two extremes). And as the reviewer suggested, in the revised manuscript, we now mention the intermediate stations and their values between Stations 3 and 9. It now reads as follows (on lines 176-181), "In line with the high N₂ fixation rates in the north, we find the lowest $\delta^{15}\text{N-PON}_{\text{Zeu}}$ signature at Station 3 (northernmost analyzed station) of -1.4 ‰. In contrast, a significantly higher $\delta^{15}\text{N-PON}_{\text{Zeu}}$ value of 1.6 ‰ was observed in the southern basin at Station 9, while intermediate values were observed for stations in between ranging from -0.8 to -0.1 ‰. Together the $\delta^{15}\text{N-PON}_{\text{Zeu}}$ north-south trend corroborates the differences in surface N₂ fixation across the lake."

On the reviewer's second point, unfortunately performing a statistical correlation between $\delta^{15}\text{N-PON}_{\text{Zeu}}$ versus N₂ fixation rates is not possible because N₂ fixation rates were not measured at exactly the same stations where

samples for $\delta^{15}\text{N-PON}$ were taken (only three stations overlap), which we have now clarified in the caption of Figure S9. Instead, if we do a correlation analysis of $\delta^{15}\text{N-PON}_{\text{Zeu}}$ against N_2 fixing cyanobacteria (where we have overlapping datasets), then we observe a significant correlation ($R^2 = 0.63$; $p < 0.05$; $N = 5$; Fig. S9f). Moreover, $\delta^{15}\text{N-PON}_{\text{Zeu}}$ correlations against the nitrate flux ($R^2 = 0.79$; $p < 0.05$; $N = 5$; Fig. S9d) and $\text{Z}_{\text{eu}} - \text{TC}$ ($R^2 = 0.61$; $p < 0.05$; $N = 5$; Fig. S9e) are statistically robust, corroborating the north-south differences in surface N_2 fixation across the lake (by proxy of the $\delta^{15}\text{N-PON}_{\text{Zeu}}$ signatures). In the revised manuscript, we have now added this information above on lines 191-193 and 209-211.

Also, calling Station 3 “the northernmost station” here is a bit confusing. I think it is the northernmost station at which $\delta^{15}\text{N-PON}$ data is available. Please rephrase accordingly. In the previous version of the manuscript, on line 177 it was already stated that “we find the lowest $\delta^{15}\text{N-PON}$ signatures at Station 3 (northernmost analyzed station)”.

Line 238: add superscript “15” to “albeit N_2 incubations were not performed...”
Done.

Line 424: using the average labeling percentage from 2018 (“mean_2018_labeling%”) for 2019 samples due to supposedly inaccurate labeling percentage data in 2019 seems like a source of error with potentially large consequence on the final rate estimate. Please provide a “sensitivity analysis” to inform the reader how impactful this error might be.

For example, the authors could calculate each 2019 N_2 -fixation rate with

a) [mean_2018_labeling%] + [1 SD]

b) [mean_2018_labeling%] - [1 SD]

where SD is the standard deviation of the 2018 labeling percentages.

Alternatively, as it seems like the 2019 N_2 -fix data are only used in Figure S6 and barely discussed in the main text, consider adding an appropriate warning/caveat (“based on uncertain labeling percentage, see Methods”) wherever the 2019 N_2 -fix data are used. As a further alternative, it is not clear to me that the 2019 N_2 -fixation rate data are really needed at all in this article and the authors might prefer not publishing them given the methodological flaw.

We agree with the reviewer that using 2018 values for data collected in 2019 would introduce a source of error. After careful consideration, we decided that the 2019 data, which is not necessary to the crux of the manuscript, be removed given the potential for a methodological flaw as the reviewer suggested. In the revised manuscript, we have removed the mention of the 2019 N_2 fixation data in both the main text and the supplementary figures/tables, thereby improving the manuscript's readability.

Line 433: I suggest rephrasing to “given that differences were small,...”
Done.

Line 456: I would ask the authors to briefly pick up this caveat (rates from anoxic zone being conservative due to trace amounts of O_2) in the Results and Discussion section, since it is common for readers to ignore M&M sections and this seems to be an important qualifier that should not get lost in the Methods section.

In the previous version of the manuscript, we have already included the caveat that rates from anoxic zone may be impacted by trace oxygen contamination on lines 244-248, “Moreover, we cannot completely exclude that

trace oxygen contamination may have inhibited N₂ fixation by strict anaerobes, such as *Chlorobium*, which dominate in the ACM (discussed below). Therefore, the rates measured in the anoxic zone likely represent the lower limit of deep/anoxic N₂ fixation activity in Lake Tanganyika.”

Figures/Table

Table 1: please indicate here and/or elsewhere which stations are counted as north/center and which as south.

We thank the reviewer for highlighting this missing information. The values indicated for the North/Center and South basins are based on Stations 1-6 and 7-9, respectively. In the revised manuscript, this is now indicated in the caption of Table 1 (lines 857-858), and in Figure 1 above panels b-m.

References 51 and 52: Please subscript the “2” in “N₂”.

Done.

Supplementary: In several cases (Schubert et al., Callbeck et al.), there is a comma behind ‘et al.’ that shouldn’t be there.

Done.

Reviewer #2 (Remarks to the Author):

Manuscript ID: NCOMMS-22-49341A

August 7, 2023

Reviewer 2

General:

I thank the authors for their careful consideration of my initial comments on the MS. I believe the article is much improved and I have only one major point on which clarification is needed (below). I have also included several minor recommendations (associated with line numbers in the revised MS) to aid the authors in polishing their article.

We thank the reviewer for their diligence and thoughtful discussions which have helped improve the manuscript.

Major comment:

In their reviewer response, the authors indicated the following: “Calculating [PN at t=0] was not possible. The natural abundance isotope samples were analyzed in different batches, and hence, the standards were not arranged in a wide mass range to derive the PN mass of the samples.” I understand that the authors cannot provide PN mass and I do not consider this to be an issue. It is very common (and generally reasonable) to assume constant PN concentration over the course of the incubation. However, in systems where denitrification occurs, it is important to measure the natural abundance of ¹⁵N-PN (i.e., the t=0 isotope ratio in the mixing model) and not to assume a value of 0.366 atom-%

(as many oceanographic studies do) because denitrification can significantly alter ^{15}N -PN natural abundance. Failing to consider this can lead to false positives. Can the authors confirm that they used natural abundance ^{15}N -PN from the same water mass from which Nfix incubation water was collected to calculate their N_2 fixation rates? (And my apologies for not clearly articulating the question last time.)

We apologize for not clarifying this in the previous version of the manuscript. Yes, we used the atom percent of the control samples as a natural abundance reference, and indeed, these control samples were taken at the same depth as the incubation experiments to determine the N_2 fixation rates. Hence, this allowed us to account for changes in the $\delta^{15}\text{N}$ -PN pool that may have been induced by other N turnover processes, such as denitrification, during the incubation experiment. In the revised manuscript, we have clarified how bulk rates of N_2 fixation were calculated, which includes the information above as well as Equation 2 (see below; and on lines 437-443):

“Bulk rates of N_2 fixation (in $\text{nmol L}^{-1} \text{d}^{-1}$) were calculated according to the following equation.

$$N_2 \text{ fixation rate} = \left(\frac{\text{at\%PON}_{\text{sample}} - \text{at\%PON}_{\text{control}}}{\text{at\%N}_2 - \text{at\%PON}_{\text{control}}} \right) \times \left(\frac{\text{mass of PON per volume}}{\text{time}} \right) \quad (\text{Eq.2})$$

Whereby the $\text{at\%PON}_{\text{sample}}$, $\text{at\%PON}_{\text{control}}$, and at\%N_2 represent the atomic $\%^{15}\text{N}$ in the particulate organic nitrogen (PON) of the incubated sample, the natural abundance of the control sample, and the N_2 pool, respectively. Notably, the control samples (used for natural abundance measurements) were sampled at the same depth as the $^{15}\text{-}^{15}\text{N}_2$ amendment experiments. The mass of PON per volume (i.e., concentration) and time of incubation were also used as inputs in Eq.2.”

Minor comments:

[22] Remove commas before and after “yet”

Done.

[35-37] I recommend placing this sentence as the first of the following paragraph, rather than having it as its own paragraph.

Done.

[38] Please replace “may” with “can”

Done.

[49] Cyanobacteria are a type of microbe.

We thank the reviewer for pointing this out, we have now dropped the word “microbial” in the line “ N_2 -fixing microbial communities”, as it was indeed redundant with “cyanobacteria”.

[52-55] Upwelling systems in the ETNP and ETSP are relatively Fe-poor and Nfix is likely limited by Fe there. Other upwelling systems (eg the Benguela, see Sohm et al 2011, or California Current System, see Kendra Turk-Kubo’s work) can have higher Nfix rates than the ocean gyres.

Yes, we agree that making a blanket statement about OMZs is difficult as their chemistry can vary. We agree with the reviewer that iron is limiting, especially if we are referring to the open ocean OMZ. However, water column iron concentrations can increase moving toward the coastal OMZ shelf as they become more influenced by benthic fluxes, according to work by Schlosser et al. 2018. To cover this nuance, in the original manuscript

(on line 52) we simply stated “...(and potentially iron)...”, to indicate that it may or may not be limiting in OMZ, and we hope this was clearly articulated.

Regarding the rates of N₂ fixation in eastern boundary upwelling systems, we thank the reviewer for providing the additional reference to N₂ fixation rates in other upwelling regions– we have incorporated references by Sohm et al. 2011 and Turk-Kubo et al. 2021 in the revised manuscript. In addition, we would like to further clarify that in the current version of the manuscript (on line 55), we indeed acknowledge that N fixation rates can vary widely in upwelling systems, “ranging from below the limit of detection to as high as 100 nM d⁻¹ (Fernandez et al. 2011; Sohm et al. 2011; Dekaezemacker et al. 2013; Luo et al. 2014; Knapp et al. 2016; Bonnet et al. 2017; Wang et al. 2019; Selden et al. 2019, 2021; Turk-Kubo et al. 2021; Kittu et al. 2023)”.

We would also like to briefly comment on the studies raised by the reviewer. In the Benguela upwelling work by Sohm et al., 2011, they observed N₂ fixation rates between 2 to 8 nmol L⁻¹ d⁻¹, which are not necessarily outliers if we compare to rates in the Eastern Tropical South/North Pacific upwellings that range from 0.4 to 3.5 nmol L⁻¹ d⁻¹ (Fernandez et al. 2011; Bonnet et al. 2013; Löscher et al. 2014; Jayakumar et al. 2017). Moreover, δ¹⁵N-PON measurements – used as a proxy for integrated N₂ fixation – find that surface values in the Benguela upwelling, are consistent with values in the other major upwelling regions (Reeder et al. 2022). Ultimately Reeder et al. 2022, conclude that N₂ fixation is low in the Benguela upwelling.

The reviewer is correct in pointing out that high rates of N₂ fixation, attaining up to 23.0 nmol L⁻¹ d⁻¹, were measured in the California Current system (Turk-Kubo et al. 2021). Interestingly, however, their study highlights that the elevated rates were “linked to post-upwelling conditions”; or in other words, when upwelling observed a cessation (at least according to their Fig. S2). Along those lines, we have also pointed out in our discussion (please see lines 365-368) that N₂ fixation has the potential to be high when more stratified conditions ensue in upwelling regions. Periodic interruptions in upwelling (i.e., leading to more stratified conditions) are commonly induced by mesoscale eddy formations, and hence, could ostensibly favor a niche for diazotrophs in ‘upwelling’ regions.

Zooming out from the specific studies above, the consensus view based on both integrated long-term indicators of N₂ fixation (δ¹⁵N-PON) and short-term incubation experiments (¹⁵-¹⁵N₂ additions) is that rates are generally low in major marine upwelling systems (Luo et al. 2014; Knapp et al. 2016; Bonnet et al. 2017; Wang et al. 2019; Löscher et al. 2020; Selden et al. 2021 (and refs therein); Reeder et al. 2022; Kittu et al. 2023). Moreover from a larger biogeochemical perspective, N₂ fixation contributes little as a ‘new’ N source in upwelling regions, contributing only ~ 1% to the export of organic N and carbon (Wang et al. 2019).

[63] I, personally, am not convinced that this is “likely” in all aquatic ecosystems (eg <https://www.nature.com/articles/ismej2014119>), but is probably true for pelagic marine systems.

As the reviewer pointed out, it is likely not the case everywhere, and hence agree that this statement seems to hold true for pelagic marine systems. We have therefore tempered this statement to read as follows (on lines 61-63), “Despite their prevalence in pelagic environments (Farnelid et al. 2013; Jayakumar and Ward 2020), heterotrophic diazotrophs may contribute to low rates (<1 nM d⁻¹) of N₂ fixation (Farnelid et al. 2013; Moisander et al. 2017).”

[160-167] What are the errors associated with these estimates?

In the revised manuscript, we have added the error values. Please see lines 160-163, where it reads as, “Averaged on a basin-scale, N₂ fixation in the upper 50 m added 300 ± 70 and 60 ± 40 μmol N m⁻² d⁻¹ to the north/central and south basins, respectively (Table 1). The estimated vertical nitrate flux into the euphotic zone amounted to 230 ± 30 and 500 ± 10 μmol N m⁻² d⁻¹ in the north/central and south basins, respectively.” It should be further noted that the values for atmospheric deposition and riverine input were obtained from Langenberg et al. 2003, and no error values were presented for these estimates by the authors.

[188] It may be worth pointing out that *Anabaenopsis* is also an established N fixer. We have now highlighted this on lines 184-186 where it now reads as, “In our parallel study addressing the phytoplankton composition, we found that these blooms were mainly comprised of known diazotrophs *Dolichospermum* (>99 %), and to a small extent *Anabaenopsis* (<1 %) (Schoffelen et al. 2018; Ehrenfels et al. 2021).”

[226-228] Filamentous cyanobacterial endosymbionts of diatoms have been found in nutrient-rich waters including river plumes (Amazon – Carpenter et al 1999; Mekong – Grosse et al 2010) and upwelling systems (<https://www.nature.com/articles/s41598-017-18006-5#Fig3>). Personally, I believe that the controls on (filamentous cyanobacterial) diazotrophs differ depending on their whether they are free-living or symbiotic. I recommend specifying “free-living” here.

We thank the reviewer for pointing this out, the word “free-living” was now added on line 229.

[254-256] I find the phrasing here a little confusing. I recommend rephrasing so that it is clear that N₂ fixation is not contributing directly to fixed N loss (which is nonsensical) but, rather, is supplying fixed N for denitrification. Also, aren't the denitrified waters being upwelled, supporting a niche for diazotrophs?

We agree with the reviewer that this statement was unclear. It now reads as (on lines 253-255), “Instead, the export of newly-fixed N from filamentous cyanobacteria in surface waters could supply up to 12% of the fixed N for anammox/denitrification in the northern oxygen-deficient zone (Table 1).” Relating to the reviewer's second question, in the more stratified northern basin upwelling is low and therefore unlikely to transport denitrified waters into the euphotic zone, so its contribution to establishing a niche for N₂ fixation would be considered negligible. This, of course, changes if we consider the southern basin where upwelling is a persistent feature, however, in these same waters, N₂ fixation is anyway low or near zero.

[258] Cyanobacteria are part of local production.

We agree that this statement was ambiguous. We were referring to cyanobacteria in surface waters, and for the sake of clarity we have dropped “(and to a very minor extent by local production)”.

[250-260] Do depth-integrated C fixation rates also correlate positively with N loss and N₂ fixation?

Carbon fixation rates were not determined in this study; therefore, we would not be able to do a correlative comparison between carbon fixation and N₂ fixation/N-loss. They were measured in the work by Callbeck et al., 2021 during our 2019 sampling campaign, which is different from the 2018 campaign discussed here. According to Callbeck et al. 2021, carbon fixation rates were active in both surface waters (up to 1 μM C d⁻¹) and in the *Chlorobium*-dominated anoxic chlorophyll maximum (up to 0.5 μM C d⁻¹) in the North basin. Since this paragraph is focused on nitrogen fixation, we have not added any additional information regarding rates of carbon fixation.

[342] P and Fe limitation was not measured in this study. I recommend rephrasing as “not drawn down” instead of “not limiting”.

We agree, and according to the reviewer's suggestion, we have changed the wording from “not limiting” to “not drawn down” on line 338.

[359] The N demand of these organisms may also be partially met by organic matter

degradation i.e., they may not need to fix as much N if its in excess in their (heterotrophic) diet.

Yes, exactly. These heterotrophic microbes have chosen a scavenging-based lifestyle that relies on obtaining organic nitrogen by investing in motility, a plethora of organic uptake systems, and metabolic degradation pathways (all of which are resource-heavy ATP sinks). Hence, an organic N scavenging lifestyle, we suspect comes as the cost of N₂ fixation (another ATP-demanding pathway). We have now clarified, on line 354, that these metabolic features of heterotrophic diazotrophs could be used to scavenge for organic nitrogen.

[366] The statement after the semi-colon is a fragment, not an independent clause. This is grammatically incorrect.

We thank the reviewer for highlighting this error, we now removed the semi-colon and divided the sentence into two (see line 363).

[365] The articles cited represent a biased sampling of upwelling systems—they are either focused on the ETNP/ETSP or are modeling papers. Other marine upwelling systems have shown evidence of significant N₂ fixation in upwelling systems (<https://www.nature.com/articles/s43705-021-00039-7> and <https://agupubs.onlinelibrary.wiley.com/doi/pdfdirect/10.1029/2005GL025569> and <https://www.nature.com/articles/s41598-017-18006-5> and <https://agupubs.onlinelibrary.wiley.com/doi/pdf/10.1029/2011GL048315>). Nfix in the ETNP and ETSP are probably iron-limited, at least in part, and do not host UCYN-A—a globally significant diazotroph which appears to be favored during upwelling (eg <https://www.frontiersin.org/articles/10.3389/fmars.2022.877562/full>)

Yes, we tend to focus on the ETSP region, but also a little bit on other major upwelling systems (ETNP, Benguela, and BOB). No N₂ fixation rates currently exist for the Arabian Sea upwelling. In our view, the ETSP region is well-studied compared to other upwelling environments where N₂ fixation rates are sparse or non-existent. Moreover, the ETSP region is also a major oxygen minimum zone (OMZ) and therefore shares some resemblances with the chemistry of Lake Tanganyika.

We appreciate the reviewer bringing to our attention other upwelling studies namely, Sohm et al. 2011 (Benguela upwelling), Turk-Kubo et al. 2021 (California Current System), Voss et al. 2006 (South China Sea), Wen et al. 2017 (Taiwan Strait), and Selden et al., 2022 (Arctic Ocean). As discussed in an earlier comment, we believe that N₂ fixation rates reported in the study by Sohm et al. 2011 are not out of line with the rates measured in other upwelling regions. Furthermore, other parameters, such as the $\delta^{15}\text{N-PON}$, suggest low N₂ fixation activity in the Benguela upwelling (Reeder et al. 2022). Turk-Kubo et al 2022 observed high N₂ fixation rates during post-upwelling conditions (discussed above).

Regarding the other upwelling studies, the reviewer is correct in pointing out the nuances of the topic. It should be said that rarely does one model ever apply across the board to all complex ecosystems. At the same time, we respect the consensus view on this matter, as a number of studies taking various approaches ($\delta^{15}\text{N-PON}$, $^{15-15}\text{N}_2$ additions, biogeochemical models) have shown that major upwelling/OMZ regions are not hotspots of N₂ fixation (Luo et al. 2014; Knapp et al. 2016; Bonnet et al. 2017; Wang et al. 2019; Löscher et al. 2020; Selden et al. 2021 (and refs therein); Reeder et al. 2022; Kittu et al. 2023). Although certainly exceptions exist (Voss et al. 2006; Wen et al. 2017; Selden et al. 2022), and to accommodate a more nuanced view we have stated that our findings may pertain to “some” upwelling regions. For example, it now reads as (on lines 362-363), “...may help to explain why N₂ fixation rates are generally low (< 5 nM d⁻¹) in some marine upwelling regions (Luo et al. 2014; Knapp et al. 2016; Bonnet et al. 2017; Wang et al. 2019; Löscher et al. 2020; Selden et al. 2021; Reeder et al. 2022; Kittu et al. 2023)”.

References:

- Bonnet, S., M. Caffin, H. Berthelot, and T. Moutin. 2017. Hot spot of N₂ fixation in the western tropical South Pacific pleads for a spatial decoupling between N₂ fixation and denitrification. *Proc. Natl. Acad. Sci. U. S. A.* **114**: E2800–E2801. doi:10.1073/pnas.1619514114
- Bonnet, S., J. Dekaezemacker, K. A. Turk-Kubo, T. Moutin, R. M. Hamersley, O. Grosso, J. P. Zehr, and D. G. Capone. 2013. Aphotic N₂ fixation in the eastern tropical South Pacific Ocean. *PLoS One* **8**: 1–14. doi:10.1371/journal.pone.0081265
- Callbeck, C. M., B. Ehrenfels, K. B. L. Baumann, B. Wehrli, and C. J. Schubert. 2021. Anoxic chlorophyll maximum enhances local organic matter remineralization and nitrogen loss in Lake Tanganyika. *Nat. Commun.* **12**. doi:10.1038/s41467-021-21115-5
- Dekaezemacker, J., S. Bonnet, O. Grosso, T. Moutin, M. Bressac, and D. G. Capone. 2013. Evidence of active dinitrogen fixation in surface waters of the eastern tropical South Pacific during El Niño and La Niña events and evaluation of its potential nutrient controls. *Global Biogeochem. Cycles* **27**: 768–779. doi:10.1002/gbc.20063
- Ehrenfels, B., M. Bartosiewicz, A. S. Mbonde, and others. 2021. Diazotrophic cyanobacteria are associated with a low nitrate resupply to surface waters in Lake Tanganyika. *Front. Environ. Sci.* **9**: 277. doi:10.3389/fenvs.2021.716765
- Farnelid, H., M. Bentzon-Tilia, A. F. Andersson, S. Bertilsson, G. Jost, M. Labrenz, K. Jürgens, and L. Riemann. 2013. Active nitrogen-fixing heterotrophic bacteria at and below the chemocline of the central Baltic Sea. *ISME J.* **7**: 1413–1423. doi:10.1038/ismej.2013.26
- Fernandez, C., L. Farías, and O. Ulloa. 2011. Nitrogen Fixation in Denitrified Marine Waters J.A. Gilbert [ed.]. *PLoS One* **6**: e20539. doi:10.1371/journal.pone.0020539
- Jayakumar, A., B. X. Chang, B. Widner, P. Bernhardt, M. R. Mulholland, and B. B. Ward. 2017. Biological nitrogen fixation in the oxygen-minimum region of the eastern tropical North Pacific ocean. *ISME J.* **11**: 2356–2367. doi:10.1038/ismej.2017.97
- Jayakumar, A., and B. B. Ward. 2020. Diversity and distribution of nitrogen fixation genes in the oxygen minimum zones of the world oceans. *Biogeosciences* **17**: 5953–5966. doi:10.5194/bg-17-5953-2020
- Kittu, L. R., A. J. Paul, M. Fernández-méndez, and M. J. Hopwood. 2023. Coastal N₂ Fixation Rates Coincide Spatially With Nitrogen Loss in the Humboldt Upwelling System off Peru *Global Biogeochemical Cycles*.doi:10.1029/2022GB007578
- Knapp, A. N., K. L. Casciotti, W. M. Berelson, M. G. Prokopenko, and D. G. Capone. 2016. Low rates of nitrogen fixation in Eastern Tropical South Pacific surface waters. *Proc. Natl. Acad. Sci. U. S. A.* **113**: 4398–4403. doi:10.1073/pnas.1515641113
- Langenberg, V. T., S. Nyamushahu, R. Roijackers, and A. A. Koelmans. 2003. External nutrient sources for Lake Tanganyika. *J. Great Lakes Res.* **29**: 169–180. doi:10.1016/S0380-1330(03)70546-2
- Löscher, C. R., T. Großkopf, F. D. Desai, and others. 2014. Facets of diazotrophy in the oxygen minimum zone waters off Peru. *ISME J.* **8**: 2180–2192. doi:10.1038/ismej.2014.71
- Löscher, C. R., W. Mohr, H. W. Bange, and D. E. Canfield. 2020. No nitrogen fixation in the Bay of Bengal? *Biogeosciences* **17**: 851–864. doi:10.5194/bg-17-851-2020
- Luo, Y. W., I. D. Lima, D. M. Karl, C. A. Deutsch, and S. C. Doney. 2014. Data-based assessment of environmental controls on global marine nitrogen fixation. *Biogeosciences* **11**: 691–708. doi:10.5194/bg-11-691-2014

- Moisander, P. H., M. Benavides, S. Bonnet, I. Berman-Frank, A. E. White, and L. Riemann. 2017. Chasing after non-cyanobacterial nitrogen fixation in marine pelagic environments. *Front. Microbiol.* **8**. doi:10.3389/fmicb.2017.01736
- Naithani, J., and E. Deleersnijder. 2004. Are there internal Kelvin waves in Lake Tanganyika? *Geophys. Res. Lett.* **31**. doi:10.1029/2003GL019156
- Reeder, C. F., D. L. Arévalo-Martínez, J. A. Carreres-Calabuig, T. Sanders, N. R. Posth, and C. R. Löscher. 2022. High Diazotrophic Diversity but Low N₂ Fixation Activity in the Northern Benguela Upwelling System Confirming the Enigma of Nitrogen Fixation in Oxygen Minimum Zone Waters. *Front. Mar. Sci.* **9**: 1–15. doi:10.3389/fmars.2022.868261
- Schlosser, C., P. Streu, M. Frank, G. Lavik, P. L. Croot, M. Dengler, and E. P. Achterberg. 2018. H₂S events in the Peruvian oxygen minimum zone facilitate enhanced dissolved Fe concentrations. *Sci. Rep.* **8**: 12642. doi:10.1038/s41598-018-30580-w
- Schoffelen, N. J., W. Mohr, T. G. Ferdelman, S. Littmann, J. Duerschlag, M. V. Zubkov, H. Ploug, and M. M. M. Kuypers. 2018. Single-cell imaging of phosphorus uptake shows that key harmful algae rely on different phosphorus sources for growth. *Sci. Rep.* **8**: 1–13. doi:10.1038/s41598-018-35310-w
- Selden, C. R., S. V. Einarsson, K. E. Lowry, K. E. Crider, R. S. Pickart, P. Lin, C. J. Ashjian, and P. D. Chappell. 2022. Coastal upwelling enhances abundance of a symbiotic diazotroph (UCYN-A) and its haptophyte host in the Arctic Ocean. *Front. Mar. Sci.* **9**: 1–8. doi:10.3389/fmars.2022.877562
- Selden, C. R., M. R. Mulholland, P. W. Bernhardt, B. Widner, A. Macías-Tapia, Q. Ji, and A. Jayakumar. 2019. Dinitrogen Fixation Across Physico-Chemical Gradients of the Eastern Tropical North Pacific Oxygen Deficient Zone. *Global Biogeochem. Cycles* **33**: 1187–1202. doi:10.1029/2019GB006242
- Selden, C. R., M. R. Mulholland, B. Widner, P. Bernhardt, and A. Jayakumar. 2021. Toward resolving disparate accounts of the extent and magnitude of nitrogen fixation in the Eastern Tropical South Pacific oxygen deficient zone. *Limnol. Oceanogr.* **66**: 1950–1960. doi:10.1002/lno.11735
- Sohm, J. A., J. A. Hilton, A. E. Noble, J. P. Zehr, M. A. Saito, and E. A. Webb. 2011. Nitrogen fixation in the South Atlantic Gyre and the Benguela Upwelling System. *Geophys. Res. Lett.* **38**: 1–6. doi:10.1029/2011GL048315
- Turk-Kubo, K. A., M. M. Mills, K. R. Arrigo, G. van Dijken, B. A. Henke, B. Stewart, S. T. Wilson, and J. P. Zehr. 2021. UCYN-A/haptophyte symbioses dominate N₂ fixation in the Southern California Current System. *ISME Commun.* **1**: 1–13. doi:10.1038/s43705-021-00039-7
- Voss, M., D. Bombar, N. Loick, and J. W. Dippner. 2006. Riverine influence on nitrogen fixation in the upwelling region off Vietnam, South China Sea. *Geophys. Res. Lett.* **33**: 1–4. doi:10.1029/2005GL025569
- Wang, W.-L. L., J. K. Moore, A. C. Martiny, and F. W. Primeau. 2019. Convergent estimates of marine nitrogen fixation. *Nature* **566**: 205–211. doi:10.1038/s41586-019-0911-2
- Wen, Z., W. Lin, R. Shen, H. Hong, S. J. Kao, and D. Shi. 2017. Nitrogen fixation in two coastal upwelling regions of the Taiwan Strait. *Sci. Rep.* **7**: 1–10. doi:10.1038/s41598-017-18006-5

Reviewer #1 (Remarks to the Author):

I would like to thank the authors for diligently addressing my concerns throughout this review process and compliment them on their work. I have no more additional comments on this manuscript at this point.

Below is a point-by-point response to the reviewers' comments, indicated in the red-type face.

REVIEWER COMMENTS

Reviewer #1 (Remarks to the Author):

I would like to thank the authors for diligently addressing my concerns throughout this review process and compliment them on their work. I have no more additional comments on this manuscript at this point.

We would again like to thank the reviewer for their support of this work, their attention to detail, and their constructive criticism.